# R1-Reward: Training Multimodal Reward Model Through Stable Reinforcement Learning

**Yi-Fan Zhang, Zhang Zhang,**[*] **Liang Wang**
Institute of Automation, Chinese Academy of Sciences (CASIA)

**Xingyu Lu, Xiao Hu**
Tsinghua University (THU)

**Chaoyou Fu**[*]
Nanjing University (NJU)

**Bin Wen,**[†] **Tianke Zhang, Changyi Liu, Kaiyu Jiang, Kaibing Chen,**
Kaiyu Tang, Haojie Ding, Jiankang Chen, Fan Yang, Tingting Gao,
Di Zhang, Guorui Zhou
KuaiShou

## Abstract

Multimodal Reward Models (MRMs) play a crucial role in enhancing the performance of Multimodal Large Language Models (MLLMs). While recent advancements have primarily focused on improving the model structure and training data of MRMs, there has been limited exploration into the effectiveness of long-term reasoning capabilities for reward modeling and how to activate these capabilities in MRMs. In this paper, we explore how Reinforcement Learning (RL) can be used to improve reward modeling. Specifically, we reformulate the reward modeling problem as a rule-based RL task. However, we observe that directly applying existing RL algorithms, such as Reinforce++, to reward modeling often leads to training instability or even collapse due to the inherent limitations of these algorithms. To address this issue, we propose the StableReinforce algorithm, which refines the training loss, advantage estimation strategy, and reward design of existing RL methods. These refinements result in more stable training dynamics and superior performance. To facilitate MRM training, we collect 200K preference data from diverse datasets. Our reward model, R1-Reward, trained using the StableReinforce algorithm on this dataset, significantly improves performance on multimodal reward modeling benchmarks. Compared to previous SOTA models, R1-Reward achieves a $8.4\%$ improvement on the VL Reward-Bench and a $14.3\%$ improvement on the Multimodal Reward Bench. Moreover, with more inference compute, R1-Reward's performance is further enhanced, highlighting the potential of RL frameworks in MRMs.

## 1 Introduction

High-quality Multimodal Reward Models (MRMs) Pu et al. (2025); Chen et al. (2024a); Wang et al. (2025a) play a crucial role in the development of Multimodal Large Language Models (MLLMs) Wang et al. (2024a); Deitke et al. (2024); Dai et al. (2024); Agrawal et al. (2024). In the training phase, from an algorithmic perspective, the MRM provides reward signals for RL Sun et al. (2023); Ouyang et al. (2022), directly influencing the stability and final outcomes of training. From a data perspective, a powerful MRM enables high-quality data filtering, improving data quality by removing noisy samples Zhang et al. (2025a); Lu et al. (2025). In the inference phase, the MRM facilitates test-time scaling strategies, such as the best-of-N strategy, to select the optimal responses Wang et al. (2025a). In the evaluation

---

[*]Corresponding Author
[†]Project Leader

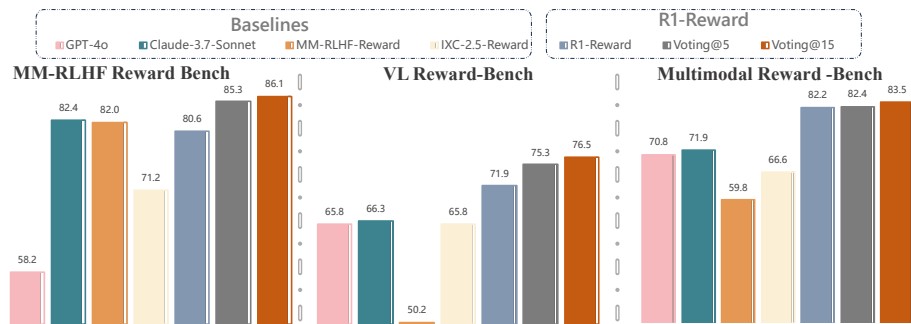

Figure 1: **R1-Reward performance on multimodal reward benchmarks.** Performance improves significantly when using a majority voting strategy (Voting@5/15) over multiple inference samples.

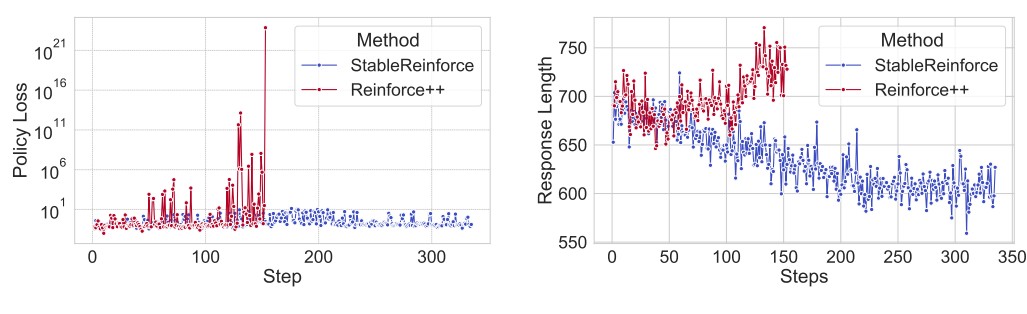

(a) Policy Loss Convergence        (b) Response Length During Training

Figure 2: **Detailed comparison between StableReinforce and Reinforce++.** (a) StableReinforce exhibits faster and more stable convergence of the policy loss during training. (b) StableReinforce continuously performs length compression, improving efficiency. Reinforce++ collapses around step 150, whereas StableReinforce remains stable, demonstrating its enhanced robustness. Additionally, after RL training with StableReinforce, the average response length is reduced by approximately 15% compared to base model, suggesting potential improvements in reasoning token efficiency.

phase, a good MRM can serve as an evaluator to simplify the evaluation process, especially in open-ended scenarios Xiong et al. (2024).

Recently, reinforcement learning DeepSeek-AI (2025) has gained widespread application in the post-training process of MLLMs Yu et al. (2025a), achieving remarkable improvements in traditional vision tasks Liu et al. (2025a); Shen et al. (2025a), multimodal reasoning tasks Huang et al. (2025); Meng et al. (2025), video understanding tasks Feng et al. (2025), and omni-tasks Zhao et al. (2025). Compared to traditional post-training strategies such as supervised fine-tuning and direct preference optimization Rafailov et al. (2023), RL offers better generalization Chu et al. (2025) and demonstrates the ability to induce long-term reasoning capabilities DeepSeek-AI (2025). However, recent improvements in MRMs have primarily focused on data Xiong et al. (2024) and structural aspects Zhang et al. (2025a), with little discussion on whether RL can be used to introduce long-term reasoning in order to improve multimodal reward modeling performance.

In this paper, we investigate whether RL algorithms can be applied to multimodal reward modeling tasks? Intuitively, the reward modeling problem can be transformed into a rule-based RL task, where the input consists of a given question and two answers. The target of the policy is to decide which answer is better. The reward during training can be obtained by comparing whether the model's judgment is consistent with the ground truth. Our goal is to enable the model to perform long-term reasoning and then provide the correct judgment. However, RL for reward modeling presents several unique challenges, and directly using traditional RL methods can easily cause training to collapse:

1. **Limitation of PPO Schulman et al. (2017) and Related Algorithms Shao et al. (2024a).** PPO and related algorithms rely on clipping the loss function to ensure training stability. However, we observe that when the advantage is negative and the current policy differs significantly from the reference policy, simple clipping fails to prevent instability, which may cause the training process to diverge or even crash.

2. **Instability of Advantage Normalization.** We observe that in the later stages of training, where the majority of rewards in a single batch are either 1 or 0 with very low variance, the commonly used advantage normalization technique (subtracting the mean and dividing by the variance) in algorithms such as GRPO Shao et al. (2024a) and Reinforce++ Hu (2025) can lead to extremely large or small advantage values for some samples. This can cause significant instability during training.

3. **Inconsistency Between Reasoning and Results.** During training, we frequently observe inconsistencies between the model's reasoning process and its final output. The model may judge one answer as better during reasoning but ultimately output an opposite conclusion. This happens because rule-based RL only scores the result without supervising the reasoning process, leading the model to learn to generate correct answers without coherent reasoning.

To this end, we propose StableReinforce, which introduces several modifications to traditional RL methods. Specifically, we refine the clipping operation to mitigate numerical instability caused by large updates and introduce a robust advantage normalization technique that limits the impact of outliers. On the reward function design front, StableReinforce introduces a novel mechanism: the use of an MLLM as a referee. This referee evaluates the consistency between the model's reasoning process and the result, ensuring that the reasoning aligns with the output. This consistency reward promotes more accurate and logically coherent decision-making. During the training phase, directly training the MLLM using reinforcement learning yields suboptimal results. Therefore, a progressive difficulty training strategy is adopted. Initially, 200K preference data is collected from publicly available datasets, and GPT-4o generates corresponding thinking processes, referred to as R1-Reward-200K, to serve as cold-start data. Meanwhile, for each sample, the number of sampling attempts GPT-4o requires to infer a conclusion matching the ground truth is recorded, which is considered the difficulty level of that sample. In the reinforcement learning phase, samples where GPT-4o requires at least two sampling attempts to arrive at the correct answer, or fails to answer correctly even after three attempts, are selected as training data. These samples are then used to train the model with the StableReinforce algorithm.

As shown in Figure 2, the RL phase effectively performs token compression, and also resulting in a noticeable performance improvement in our experiments. R1-Reward performs excellently on common multimodal reward modeling benchmarks. As shown in Figure 1, R1-Reward outperforms the state-of-the-art (SOTA) on all the three benchmarks. Furthermore, R1-Reward exhibits strong inference time scalability. By sampling only five times and selecting the most frequent answer as the correct one, the accuracy of reward modeling improves substantially. On the MM-RLHF Reward Bench Zhang et al. (2025a), VL Reward-Bench Li et al. (2024a), and Multimodal Reward Bench Yasunaga et al. (2025), R1-Reward achieves improvements of $3.5\%$, $13.5\%$, and $14.6\%$, respectively, compared to SOTA. Finally, downstream validation shows consistent improvements across multiple benchmarks when using R1-Reward to train smaller MLLMs, confirming the practical utility and transferability of our reward modeling approach.

## 2 Preliminary and Limitations

**Standard Reward Models.** Typically, a reward model starts with an LLM $\phi$, where the LLM head $h_l$ is replaced with a linear reward head $l_r$, enabling the model to output a scalar reward. These models are trained using human-provided pairwise comparisons. Given a query $\mathbf{x}$, a preferred and a less preferred response $y_w, y_l$, the reward model is optimized to assign higher rewards to preferred responses: $\ell_{\text{Reward}}(\theta) = \mathbb{E}_{\mathbf{x}, y_w, y_l} \left[ -\log \sigma \left( r(y_w|\mathbf{x}) - r(y_l|\mathbf{x}) \right) \right]$, where $r(y|\mathbf{x})$ is the reward and $\sigma$ is the sigmoid function.

**Proximal Policy Optimization (PPO)** Schulman et al. (2017) is a commonly used algorithm in RL that aims to optimize a policy directly while maintaining stable and efficient learning. PPO belongs to the family of policy gradient methods, where the objective is to improve the policy by maximizing the expected cumulative reward. Unlike traditional policy gradient methods, which can suffer from large updates and instability, PPO introduces a novel way to constrain policy updates, ensuring both efficient and stable learning. The objective function for PPO is defined as: $L^{\text{PPO}}(\theta) = \frac{1}{|t|} \sum_t \left[ \min \left( \frac{\pi_\theta(a_t|s_t)}{\pi_{\theta_{\text{old}}}(a_t|s_t)} A_t, \text{clip} \left( \frac{\pi_\theta(a_t|s_t)}{\pi_{\theta_{\text{old}}}(a_t|s_t)}, 1-\epsilon, 1+\epsilon \right) A_t \right) \right]$, where $\pi_\theta(a_t|s_t)$ is the probability of taking action $a_t$ at state $s_t$ under the current policy $\theta$; $\pi_{\theta_{\text{old}}}(a_t|s_t)$ is the probability under the old policy with parameters $\theta_{\text{old}}$; $A_t$ is the advantage estimate at time $t$, which measures the relative desirability of the action taken; and $\epsilon$ is a small hyperparameter (typically $0.1 \leq \epsilon \leq 0.3$) that controls how much the policy can change.

The first term in the minimum represents the standard objective, while the second term applies a clipping mechanism. The clip function restricts the ratio of the new policy to the old policy to stay within the interval $[1-\epsilon, 1+\epsilon]$. If the ratio exceeds this range, the objective is capped, preventing large updates that could destabilize the learning process.

**Reinforce++ Hu (2025) Enhancements.** Reinforce++ incorporates several key optimizations to enhance training stability and efficiency of PPO. One is the addition of a token-level Kullback-Leibler (KL) divergence penalty between the RL model and the supervised fine-tuning (SFT) model distributions. Additionally, Reinforce++ introduces global batch-level reward normalization, clipping, and scaling for stability, as well as advantage normalization: $A_{\text{normalized}} = \frac{A - \mu_A}{\sigma_A}$ Where $\mu_A$ and $\sigma_A$ are the mean and standard deviation of the advantage values. REINFORCE++ is shown to be more stable compared to GRPO Shao et al. (2024b) and faster than PPO Xie et al. (2025a); Cui et al. (2025).

## 2.1 Drawbacks of Traditional PPO/Reinforce++

During our training process, we observed two core issues in the Reinforce++ algorithms that can easily lead to model instability and poor performance, especially for reward model training.

**Instability Caused by Training Losses.** The typical PPO loss function is implemented as follows, given the log probabilities $\log \pi_\theta(a_t|s_t)$, $\log \pi_{\theta_{\text{old}}}(a_t|s_t)$, and advantages. The pseudocode in for calculating the loss is shown in Algorithm 1 (lines 0-3). If the ratio $\frac{\pi_\theta(a_t|s_t)}{\pi_{\theta_{\text{old}}}(a_t|s_t)}$ differs significantly, two main issues arise. First, the expression $(\log\_probs - \text{old}\_\log\_probs) . \exp()$ can lead to numerical instability. When the difference in token probabilities is large, the exponential function may overflow, causing the model to crash. Even if the computation proceeds normally, if the advantage is negative, $-\text{torch.min}(\text{surr1}, \text{surr2})$ could result in an excessively large loss due to the minimization objective. For example: let $\log\_probs = [-0.1, -0.1, -0.1, -0.1]$, $\text{old}\_\log\_probs = [-10, -0.2, -0.2, -5]$, and advantages $= [-1.0, -1.0, 0.5, -0.5]$, the resulting loss values might be: loss $= [19930.4, 1.1, -0.5, 67.1]$. Such large losses can make the optimization process highly unstable. Currently, many training methods remove the KL divergence constraint Meng et al. (2025); Peng et al. (2025), allowing each mini-batch to perform multiple parameter updates, thereby improving data usage efficiency Hu (2025); Schulman et al. (2017). The former accelerates model updates, while the latter further increases the discrepancy between $\log \pi_\theta(a_t|s_t)$ and $\log \pi_{\theta_{\text{old}}}(a_t|s_t)$. Consequently, in these cases, the ratio between these two values can diverge significantly, leading to instability.

**Instability Caused by Advantage Normalization.** In addition to the training loss, the data labels for the reward model are relatively simple, consisting of only two labels: 1 and 2, which makes them easy to learn. As a result, during training, there is a high probability that the majority of the batch will correctly predict the rewards. In extreme cases, such as a batch containing 255 rewards of 1 and 1 reward of 0, this highly imbalanced distribution, when subjected to z-Normalization, can lead to significant numerical disparities. Particularly, the advantage corresponding to the 0 reward in this example would be normalized to -15.96. A large advantage value like this can cause instability.

---

**Algorithm 1** Pseudocode of PPO Loss Function in a PyTorch-like style.

---

```
# log_probs: log probabilities of the current policy
# old_log_probs: log probabilities of the previous policy
# advantages: advantage estimates for the actions
# epsilon: clipping parameter for PPO objective

# Our Pre-Clip strategy: clip the log difference to prevent large values
log_diff = log_probs - old_log_probs
log_diff = torch.clamp(log_diff, max=np.log(1e3), min=np.log(1e-3)) # similar to 10
ratio = torch.exp(log_diff)

# PPO strategy
0. ratio = (log_probs - old_log_probs).exp() # compute the probability ratio

1. surr1 = ratio * advantages # first surrogate objective
2. surr2 = ratio.clamp(1 - epsilon, 1 + epsilon) * advantages # second surrogate with
    clipping

# The final loss is the minimum of the two surrogates
3. loss = -torch.min(surr1, surr2) # negative loss for minimization
```

---

## 3 R1-Reward

### 3.1 OUR TRAINING ALGORITHM: STABLEREINFORCE

To overcome the drawbacks and enhance the stability of reinforcement learning training, we propose two strategies: pre-CLIP and advantage filter, which respectively remove unstable gradients and advantages that deviate excessively from the overall distribution. In terms of reward design, we introduce the consistency reward to ensure consistency between reasoning and the final answer.

**Pre-CLIP.** As shown in Algorithm 1 under the "Our Pre-Clip strategy", our core approach is to clip large ratios before computing the exponential of the log probability. The value of $1e3$ is a hyperparameter that we find works well and the method is relatively insensitive to hyperparameter variations. The main purpose of this step is to mitigate the impact of noisy data on the overall training process with log-probability clamping:

$$\overline{\frac{\pi_\theta(a_t|s_t)}{\pi_{\theta_{\text{old}}}(a_t|s_t)}} \leftarrow \exp\left(\text{clip}\left(\log\frac{\pi_\theta}{\pi_{\theta_{\text{old}}}}, \log\delta_{\min}, \log\delta_{\max}\right)\right)$$

where $\delta_{\min} = 10^{-3}$, $\delta_{\max} = 10^3$ control allowable probability ratio bounds. By clipping the ratio before applying the exponential function, we can prevent overflow issues due to excessively large differences in the ratios. Additionally, this clipping ensures that large log-probability differences are mitigated, particularly when the advantage is negative, thus maintaining training stability.

**Advantage Filter.** To prevent the influence of outliers due to the extreme imbalance in the advantage distribution, we apply the 3-sigma rule. For the standardized advantage, $A_{\text{standardized}} = \frac{A - \mu_A}{\sigma_A}$, we retain only those advantages that fall within the range of $[-3, 3]$[1]. This range corresponds to values within 3 standard deviations from the mean in the original distribution, as the standardization process converts the data to z-scores (unitless measures in terms of standard deviations). In the extreme case from the previous subsection, this ensures that all samples with original rewards of 1 are selected, while extreme negative advantages are excluded.

$$\hat{A} = \begin{cases} A_{\text{standardized}} & \text{if } |A_{\text{standardized}}| \leq 3 \\ 0 & \text{otherwise} \end{cases}, \quad A_{\text{standardized}} = \frac{A - \mu_A}{\sigma_A + \epsilon}$$

The final StableReinforce objective function with clipping applied:

---

[1]After applying Z-normalization in the original text, the distribution becomes a standard normal distribution, meaning it has a mean of 0 and a standard deviation of 1.

Table 1: **Prompt template for reward model training.**

---

You are a highly skilled and impartial evaluator tasked with comparing two responses generated by a Large Multimodal Model for a given question.

- Start with a thorough, side-by-side comparative analysis enclosed within `<think>` and `</think>` tags. A tie is not permitted; you must choose a better option.

- Conclude with a single numeric choice enclosed within `<answer>` and `</answer>` tags:

- Output "1" if Response 1 is better; Output "2" if Response 2 is better.

**Input**

```
[Question]: {question};[Response 1]: {answer1};[Response 2]:
{answer2}
```

**Output Format (strictly follow)**

```
<think>Your detailed comparative
analysis</think><answer>1/2</answer>
```

---

$$L^{\text{StableReinforce}}(\theta) = \frac{1}{|t|} \sum_t \left[ \min \left( \overline{\frac{\pi_\theta(a_t|s_t)}{\pi_{\theta_{\text{old}}}(a_t|s_t)}} \hat{A}_t, \text{clip} \left( \overline{\frac{\pi_\theta(a_t|s_t)}{\pi_{\theta_{\text{old}}}(a_t|s_t)}}, 1 - \epsilon, 1 + \epsilon \right) \hat{A}_t \right) \right],$$

We discuss the difference between StableReinforce and recent RL methods for LLMs in section B.

### 3.2 REWARD FUNCTION AND TRAINING DATA

Inspired by DeepSeek-R1 DeepSeek-AI (2025), we aim to directly use RL to guide the reward model in generating the best analysis content, in order to produce high-quality model output comparisons. As a result, the prompt format in Table 1 transforms the reward modeling task into a straightforward rule-based reinforcement learning problem. By defining the model's output format, we only need to define our reward functions to complete the training process:

- **Formatting Reward.** The model's output must adhere to a specific format of '<think> </think><answer> </answer>', which encourages the model to reason before generating the final output. This ensures that the model reflects on the reasoning process before making its final decision, enhancing both the quality and interpretability of the generated content.

- **Result Reward.** The model's generated final result must align with human preferences. This primarily involves ensuring that the model's output ranking labels are consistent with those of human experts, enhancing the overall usefulness and credibility.

**Inconsistency Between Reasoning and Results.** However, simply following existing work DeepSeek-AI (2025); Yu et al. (2025b) in our setting has led to unexpected results. During training, we observe discrepancies between the model's reasoning and its final answer. For example, the reasoning might conclude that response 2 is better but the model outputs answer 1, as seen in *<think>... response 2 is better</think><answer>1</answer>*. This inconsistency arises because, we provide no supervision for the reasoning process and only score based on the outcome. When a sample demonstrates poor reasoning but produces the correct answer, this pattern is inadvertently reinforced, leading the model to believe that reasoning and the final answer are not necessarily linked. Consequently, the model may learn to generate correct answers without a coherent reasoning process. This could even result in the model treating the reasoning process as irrelevant, or worse, outputting repetitive content or random noise. To address this issue, we introduce an additional component, Qwen2.5-VL-7B-Instruct, as a supervisor to verify whether the reasoning and the final result are consistent. This addition helps ensure that the reasoning process and output align well, introducing the following reward function:

Table 2: **Summary of datasets used for training**, including the category (text or image), dataset name, the number of original samples, and the number of samples selected for final training.

| Dataset | # Original | # Sample | Dataset | # Original | # Sample |
|---|---|---|---|---|---|
| RLAIF-V Yu et al. (2024a) | 74,802 | | MM-RLHF-Long Zhang et al. (2025a) | 41,163 | 41,163 |
| VL-Feedback Li et al. (2023) | 80,258 | 100k | MM-RLHF-Short Zhang et al. (2025a) | 46,281 | 46,281 |
| POVID Zhou et al. (2024) | 17,184 | | MM-RLHF-Mcq Zhang et al. (2025a) | 8,306 | 8,306 |
| WildVision-Battle Lu et al. (2024a) | 10,383 | | MM-RLHF-Safety Zhang et al. (2025a) | 9,990 | 9,990 |

- **Consistency Reward.** The model's final result must be consistent with its intermediate reasoning process. This function ensures that the final answer is directly derived from the model's reasoning process, rather than being generated in isolation from the reasoning steps.

Integrating the consistency reward as a separate reward and combining it with the previous two reward functions can lead to a situation where the model, despite selecting the wrong answer, may still receive a high overall reward due to the consistency component. This could result in the model overly prioritizing consistency. To mitigate this issue, the final reward is designed as follows:

$$\text{Final Reward} = \text{Result Reward} \times (1 + 0.5 \times \text{Consistency Reward}) + 0.5 \times \text{Formatting Reward}.$$

This ensures that the consistency reward is only taken into account when the result is correct, thereby preventing the model from excessively favoring consistency in cases where the outcome is incorrect.

**Dataset Construction.** As shown in Table 2, we sample preference data from multiple existing datasets for training. To ensure data quality and diversity, we sample all instances from the human-annotated dataset MM-RLHF, and an additional 100,000 samples from other multimodal preference datasets. The final dataset is termed R1-Reward-200k, which combines these diverse instances to create a robust training foundation for our model. We then randomly shuffle the data to ensure a balanced ratio of answers 1 and 2 (1:1), preventing the model from favoring a specific answer. Each sample consists of a quadruple: (question, answer 1, answer 2, ground truth choice).

**Long-CoT (Chain-of-Thought) Wei et al. (2022) Cold Start.** Since MLLMs are not initially trained for reward modeling tasks, directly using MLLMs for reinforcement learning training yields poor and unstable results. Therefore, we first use GPT-4o as an annotator for each sample in the R1-Reward-200k dataset, utilizing the prompts from Table 1 to construct SFT data. The temperature is set to 0, with a maximum of 3 attempts. We also record how many attempts GPT-4o needs to generate a final response that is judged to be correct (i.e., the same to the ground truth choice). This SFT training phase teaches the model the basic format and familiarizes it with the reward modeling task.

**RL Training Data.** We use all samples from the SFT phase with at least 2 attempts, as well as samples where GPT-4o fails to produce the correct answer after three attempts. These samples exhibit smaller differences between answer 1 and answer 2, making them more difficult.

# 4 Experiments

**Implementation Details.** Both SFT and RL experiments are conducted on $4 \times$H800 (80G) GPUs. The SFT phase trains for 1 epoch and takes approximately 8 hours, while the RL phase trains for 5 epochs and takes 12 hours. We use QwenVL-2.5-7B-Instruct as the base model for training. During the SFT phase, the learning rate is set to 1e-5, and the batch size is set to 256. We use the OpenRLHF Hu et al. (2024) framework for RL. The training batch size is set to 128, and the rollout batch size is set to 256. The learning rate is set to 1e-6, and the initial KL coefficient is set to 0.

**Baseline Algorithm.** At the algorithmic level, we primarily compare two entities: the reward model and MM-RLHF-Reward Zhang et al. (2025a). For the former, we replace the language

Table 3: **VLReward Bench.** Performance comparison of our reward model (R1-Reward) with existing open-source and private counterparts.

| Models | #Param | General | Hallucination | Reasoning | Overall Acc | Macro Acc |
|---|---|---|---|---|---|---|
| Proprietary Models | | | | | | |
| Claude-3.5-Sonnet (2024-06-22) | - | 43.40 | 55.00 | 62.30 | 55.30 | 53.60 |
| GPT-4o (2024-08-06) | - | 49.10 | 67.60 | 70.50 | 65.80 | 62.40 |
| Gemini-1.5-Pro (2024-09-24) | - | 50.80 | 72.50 | 64.20 | 67.20 | 62.50 |
| Claude-3.7-Sonnet | - | 68.08 | 70.70 | 60.81 | 66.31 | 66.53 |
| Open-Source Models | | | | | | |
| VITA-1.5 Fu et al. (2025) | 7B | 18.55 | 8.93 | 22.11 | 16.48 | 16.53 |
| SliME Zhang et al. (2024a) | 7B | 7.23 | 27.09 | 18.60 | 19.04 | 17.64 |
| InternVL2 Chen et al. (2023) | 8B | 35.60 | 41.10 | 59.00 | 44.50 | 45.20 |
| LLaVA-Critic Xiong et al. (2024) | 8B | 54.60 | 38.30 | 59.10 | 41.20 | 44.00 |
| Molmo Deitke et al. (2024) | 72B | 33.90 | 42.30 | 54.90 | 44.10 | 43.70 |
| Qwen2-VL Wang et al. (2024a) | 72B | 38.10 | 32.80 | 58.00 | 39.50 | 43.00 |
| NVLM-D Dai et al. (2024) | 72B | 38.90 | 31.60 | 62.00 | 40.10 | 44.10 |
| Llama-3.2 Minghao Yang (2024) | 90B | 42.60 | 57.30 | 61.70 | 56.20 | 53.90 |
| Reward Models | | | | | | |
| MM-RLHF-Reward Zhang et al. (2025a) | 7B | 45.04 | 50.45 | 57.55 | 50.15 | 51.01 |
| IXC-2.5-Reward Zang et al. (2025) | 7B | **84.70** | 62.50 | 62.90 | 65.80 | 70.00 |
| Ours | | | | | | |
| R1-Reward | 7B | 63.84 | 85.71 | 64.78 | 71.92 | 71.44 |
| *Voting@15* | 7B | 66.32 | **89.06** | **73.70** | **76.46** | **76.36** |

head of the base LLM with a two-layer MLP that outputs a float value as the reward. Training is done using a binary classification loss. For the latter, in addition to the traditional binary classification loss, an additional critic loss is required. Specifically, the model first outputs an evaluation of the candidate, and then, based on the evaluation, the reward head provides the reward value.

**Evaluation Benchmarks and Metrics.** The multimodal benchmark includes VL-Reward Bench Li et al. (2024a), Multimodal RewardBench Yasunaga et al. (2025), and MM-RLHF-Reward Bench Zhang et al. (2025a). VL-Reward Bench uses two metrics: *Overall Accuracy*, which measures the percentage of model decisions aligning with human preferences, and *Macro Average Accuracy*, which calculates the average accuracy across task categories to address distribution imbalance. Multimodal RewardBench evaluates reward models in six areas: correctness, preference, knowledge, reasoning, safety, and visual question answering (VQA). MM-RLHF-Reward Bench has two metrics: 1. *Acc*, measuring correct identification of the preferred response, and 2. *Acc+*, which assesses the model's ability to correctly rank all response pairs, focusing on challenging cases with small ranking differences or hard-to-distinguish pairs.

### 4.1 Main Results

We evaluate the performance of R1-Reward on three common multimodal reward model benchmarks. On the VLReward Bench (Table 3), R1-Reward achieves the best overall performance, with an average accuracy of 71.92%. This represents a roughly 9.3% improvement in overall accuracy compared to the previous best open-source model, IXC-2.5-Reward. Notably, IXC-2.5-Reward trains on more than 1 million samples, while our training data consists of 200k samples, highlighting a significant improvement in data efficiency. In comparison to other open-source models, R1-Reward demonstrates a larger margin of improvement. Among closed-source models, Gemini-1.5-Pro performs the best, but R1-Reward outperforms it across all dimensions, further demonstrating its superiority.

On the Multimodal Reward Bench (Table 4), R1-Reward achieves the best performance across all dimensions, with a 14.3% improvement over the previous state-of-the-art. It is worth noting that the Multimodal Reward Bench is derived from over ten existing benchmarks and reconstructed into a unified set, with minimal overlap with our training data. This further demonstrates R1-Reward's remarkable generalization ability across different datasets. The performance of R1-Reward on the MM-RLHF-Reward Bench and Test-Time Scaling analysis is shown in Section. C.4 and C.5.

Table 4: **Multimodal Reward Bench.** Performance comparison of our reward model (R1-Reward) with existing open-source and proprietary counterparts.

| Model | #Param | Overall | General | | Knowledge | Reasoning | | Safety | VQA |
|---|---|---|---|---|---|---|---|---|---|
| | | | Correctness | Preference | | Math | Coding | | |
| Proprietary Models | | | | | | | | | |
| GPT-4o | - | 70.8 | 62.6 | 69.0 | 72.0 | 67.6 | 62.1 | 74.8 | **87.2** |
| Gemini 1.5 Pro | - | 71.9 | 63.5 | 67.7 | 66.3 | 68.9 | 55.5 | 94.5 | **87.2** |
| Claude 3.5 Sonnet | - | 71.5 | 62.6 | 67.8 | 73.9 | 68.6 | 65.1 | 76.8 | 85.6 |
| Claude 3.7 Sonnet | - | 71.9 | 58.4 | 60.7 | **78.1** | 76.3 | 71.3 | 72.0 | 86.8 |
| Open-Source Models | | | | | | | | | |
| SliME Zhang et al. (2024a) | 8B | 42.0 | 42.3 | 52.2 | 47.5 | 43.5 | 35.3 | 19.1 | 53.8 |
| VITA-1.5 Fu et al. (2025) | 7B | 53.6 | 55.6 | 54.3 | 52.5 | 51.9 | 52.8 | 58.1 | 50.0 |
| Llama-3.2-Vision-Instruct Minghao Yang (2024) | 11B | 51.2 | 57.8 | 65.8 | 55.5 | 50.6 | 51.7 | 20.9 | 55.8 |
| Molmo-D-0924 Deitke et al. (2024) | 7B | 52.9 | 56.8 | 59.4 | 54.6 | 50.7 | 53.4 | 34.8 | 60.3 |
| Llama-3.2 Minghao Yang (2024) | 90B | 61.2 | 60.0 | 68.4 | 61.2 | 56.3 | 53.1 | 52.0 | 77.1 |
| InternVL-3 Zhu et al. (2025) | 8B | 63.6 | 59.6 | 61.6 | 60.5 | 65.1 | 56.6 | 59.3 | 82.3 |
| Qwen-2-VL Wang et al. (2024a) | 72B | 70.9 | 56.4 | 62.3 | 70.2 | 73.3 | 58.9 | 90.1 | 85.3 |
| Reward Models | | | | | | | | | |
| MM-RLHF-Reward Zhang et al. (2025a) | 7B | 67.1 | 61.7 | 67.5 | 54.3 | 58.4 | 57.9 | 92.9 | 76.8 |
| IXC-2.5-Reward Zang et al. (2025) | 7B | 66.6 | 60.7 | 64.2 | 56.8 | 63.0 | 50.5 | 89.9 | 81.1 |
| Ours | | | | | | | | | |
| R1-Reward | 7B | 82.2 | 77.5 | 74.0 | 74.9 | 83.1 | 79.6 | 99.6 | 86.5 |
| Voting@15 | 7B | **83.3** | **78.0** | **77.2** | 74.6 | 81.3 | **85.8** | 99.4 | **87.0** |

## 4.2 ABLATIONS AND ANALYSIS

**R1-Reward Demonstrates High Data Efficiency.** In Table 5, we compare the performance of a traditional reward model (using a two-layer MLP as the reward head) and MM-RLHF-Reward (which first generates a critic and then generates the reward) trained on the same dataset. For MM-RLHF-Reward, the training data must include an evaluation for each response. To achieve this, we use GPT-4o to generate corresponding evaluations for each sample, which may be slightly less accurate than the human annotations used in the original work. All the models' backbones are the same. As shown in the table, the traditional reward model, when trained with only 200K data samples, performs poorly. In most cases, MM-RLHF outperforms the traditional reward model. Its superior performance in the "hallucination" dimension is likely due to the generated critic. Comparing these two baselines, the reinforcement learning-based approach significantly enhances the reward modeling capabilities, even with the same amount of data. Moreover, our SFT approach shows advantages over both the traditional reward model and MM-RLHF-Reward. We believe this is primarily due to that we allow direct comparison of two responses during the scoring process, whereas existing methods score responses independently before comparing them.

Based on the appendix experiments, our additional studies provide comprehensive validation of the R1-Reward model and StableReinforce algorithm across multiple dimensions. The appendix demonstrates that each component of StableReinforce (advantage filter, pre-clip, and consistency reward) is essential for training stability, with ablations showing that removing any component leads to decreased performance and potential training collapse. Hyperparameter analysis confirms the robustness of our design choices. The model shows strong generalization capabilities, achieving state-of-the-art performance on the challenging MM-RLHF-Reward Bench and demonstrating effective test-time scaling where performance consistently improves with increased sampling. Human evaluation validates that our approach produces superior reasoning quality, with annotators preferring R1-Reward's reasoning processes in 72.5% of cases. The method proves robust to annotation quality, maintaining strong performance even when using weaker models like Qwen2.5-VL-7B instead of GPT-4o for data construction. Finally, downstream validation shows consistent improvements across multiple benchmarks when using R1-Reward to train smaller MLLMs, confirming the practical utility and transferability of our reward modeling approach.

## 5 Conclusion

In this paper, we introduce R1-Reward, a MRM trained using the StableReinforce algorithm. We demonstrate that RL can be effectively applied to reward modeling, significantly enhancing its performance. Our approach addresses key challenges, including training instability, the advantage normalization limitation, and inconsistencies between reasoning and results. By incorporating techniques such as pre-clipping, advantage filtering, consistency reward and a a progressive difficulty training strategy, StableReinforce stabilizes training

and improves model performance. Experiments show that R1-Reward outperforms SOTA models on several multimodal reward model benchmarks, with significant improvements in accuracy and data efficiency. While R1-Reward demonstrates excellent test-time scaling capabilities and lays the foundation for future research on incorporating reinforcement learning into MRMs, we acknowledge the limitations of our current work. Specifically, we only test a basic majority voting strategy for test-time scaling, and more advanced methods could potentially yield further performance improvements Liu et al. (2025b). Additionally, the performance of R1-Reward on the MM-RLHF Reward Bench shows only modest improvements over previous SOTA models. Therefore, improving training strategies to further enhance the foundational capabilities of reward models remains an important open problem.

## LLM Usage Declaration

In this research, LLMs were used exclusively for grammar checking and to assist with the clarity of language. No LLM was involved in the ideation or content generation processes. The authors take full responsibility for all content presented in the paper, including any generated by the LLM. We have ensured that the use of LLMs complies with ethical standards and does not constitute any form of scientific misconduct or plagiarism.

## Acknowledgement

This work was jointly supported by the National Key R&D Program of China (Grant No. 2022ZD0117901), the National Natural Science Foundation of China (Grant Nos. 62506158, 62441234, 62322607, 62276261, 62373355, 62306311, and 62236010), the Basic Research Program of Jiangsu (Grant No. BK20251183), and the Beijing Natural Science Foundation (Grant No. L252033).

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

## A    Related Work

**MLLMs.** Thanks to the success of language models, MLLMs have rapidly developed in recent years, with their task handling capabilities and model performance advancing at a fast pace Fu et al. (2024a); Zhang et al. (2024b); Yu et al. (2025a); Li et al. (2022); Zhang et al. (2025b); Yang et al. (2025a). For example, traditional multi-modal large models perform well in handling complex high-resolution images and human dialogue Bai et al. (2025); OpenAI. (2024); Li et al. (2024b); Team (2024a); Wang et al. (2024a). A series of works focus on improving the context length Shen et al. (2025b), computational efficiency Zhang et al. (2024c; 2025c), reducing hallucinations Lu et al. (2025); Zhang et al. (2024d), enhancing conversational abilities Xiong et al. (2024), and aligning with human preferences Zhang et al. (2025a). Omni-MLLMs are capable of simultaneously processing multiple modalities such as speech, video, images Li et al. (2025a); Zhao et al. (2025), and even interacting with users via voice Fu et al. (2024b; 2025). Unify-MLLMs can perform mixed-modal generation Xie et al. (2024); Team (2024b); Xie et al. (2025b); Li et al. (2025b); Zhang et al. (2025d); Yang et al. (2025b), for example, generating an image with auxiliary lines while understanding a math problem, enhancing both generation and comprehension abilities. Recently, with the success of Open AI's O1 model and Deepseek's R1 model DeepSeek-AI (2025), the rule-based reinforcement learning approach has gained significant attention in the multi-modal field. At the same time, "think with image" paradigm OpenAI (2025); Zhang et al. (2025e) has also attained much attention for visual reasoning. Various studies are devoted to enhancing the reasoning capabilities of multi-modal models. However, as far as we know, no work has yet explored whether the reinforcement learning paradigm can be transferred into reward modeling.

**Reward Model Training.** The reward models most relevant to this paper are pure text reward models and multi-modal reward models. There are generally three main approaches to reward modeling. The first approach is to directly use a language model or multi-modal model as the reward model by designing precise prompts that allow them to output a score or ranking Xiong et al. (2024); Zhang et al. (2025f). However, this method heavily depends on the model's instruction-following ability and comprehension. The second approach involves connecting the latent representation of a language model to a reward head (typically an MLP or linear layer), where the model directly outputs a score. During training, the reward modeling is converted into a binary classification task. This approach is computationally efficient, but it lacks interpretability Liu et al. (2024a); Zang et al. (2025); Minghao Yang (2024); Lou et al. (2024); Wang et al. (2024b). The final type of model simultaneously learns to evaluate the question-answer pair and creates an additional reward head to provide the score Yu et al. (2024b); Zhang et al. (2025a). This model strikes a balance between interpretability and computational efficiency, but it usually requires specific data formats or training strategies. This paper proposes training a reward model through reinforcement learning. The model first outputs an inference for a given question-answer pair and ultimately provides a ranking. Through reinforcement learning, we force the model to learn the format of the reward modeling task, avoiding the shortcomings of the first approach without requiring an additional reward head, while maintaining the model's interpretability.

## B    Comparison to Existing RL Studies

In the field of RL for LLMs, recent concurrent advancements have emerged, some of which share similarities with our approach or report analogous observations. Although these methods have not been directly applied to multimodal domains or reward modeling, we provide a concise discussion in this section for comparative purposes. Notably, DAPO Yu et al. (2025b), TOPR Roux et al. (2025), and Minimax-01 Li et al. (2025c) focus on improving CLIP operations, particularly in the design of the epsilon parameter. In contrast, our approach fundamentally differs by clipping the logits ratio prior to the exponential operation. This strategy enhances numerical stability and mitigates the adverse effects of negative advantages. Similarly, Dr. GRPO Liu et al. (2025c) identifies the detrimental impact of advantage normalization and adopts a strategy of setting variance to 1. However, in scenarios with high

Table 5: **Evaluation results** on VL Reward Bench comparing different models and training setups, including baselines, models trained with R1-Reward-200K, and ablation studies (`Ours`).

| # Data | Models | VL-Reward Bench | | | |
|---|---|---|---|---|---|
| | | **General** | **Hallucination** | **Reasoning** | **Overall Acc** |
| | | Baselines | | | |
| More than 1M | IXC-2.5-Reward | 84.70 | 62.50 | 62.90 | 65.80 |
| MM-RLHF-120K | MM-RLHF-Reward | 45.04 | 50.45 | 57.55 | 50.15 |
| | | Trained by R1-Reward-200K | | | |
| R1-Reward-200K | Reward Model | 56.71 | 56.03 | 48.67 | 56.41 |
| R1-Reward-200K | MM-RLHF-Reward | 61.01 | 62.28 | 59.30 | 60.80 |
| | | Ours | | | |
| R1-Reward-200K | StableReinforce | 63.84 | 85.71 | 64.78 | 71.92 |
| R1-Reward-200K | wo advantage | 63.43 | 77.45 | 62.38 | 68.96 |
| R1-Reward-200K | wo pre-clip | 62.06 | 77.44 | 61.23 | 67.36 |
| R1-Reward-200K | Reinforce++ | | Collapse | | |
| R1-Reward-200K | Only Long-Cot SFT | 59.92 | 72.27 | 60.01 | 64.80 |

original variance, this approach allows extreme values to dominate. Instead, we employ a 3-sigma filter, which preserves the benefits of z-normalization while effectively removing outliers.

# C    Additional Experiments

## C.1    Ablation Studies of the StableReinforce Algorithm.

We examine the impact of each component of the StableReinforce algorithm on the training process and final results. First, we emphasize the necessity of the Consistency Reward Function. Removing this function results in significant hallucination behaviors across different algorithms, making it challenging to achieve stable evaluation outcomes. Additionally, directly applying the Reinforce++ algorithm causes the model to crash, with the loss becoming NaN and the response length reaching the preset maximum length, while the output consists entirely of garbled text. In Table 5 and Figure 4, we present the effects of removing each module on final performance and changes in training dynamics. We observe that the Advantage Filter and Pre-Clip modules primarily ensure training stability by effectively removing outliers from the loss. Removing any of these components results in decreased final accuracy, reduced training stability, and the model's output length failing to converge to shorter values.

## C.2    Ablation Studies of Hyperparameters.

Regarding hyperparameters, the advantage filter directly follows the 3-sigma rule, a common statistical screening strategy, with no extra hyperparameters. The pre-clip parameter is set to np.log($10^3$) as an empirical value. This corresponds to a policy update ratio exceeding 10, which is an extremely drastic update. Larger updates are usually not effective learning signals but rather caused by instability or noise in early training. We conducted an ablation study on this value, with partial results shown in Table 6. We observe that setting pre-clip between 5-10 results in minor impact, but if set very large (e.g., 100), performance degenerates to nearly the same as without pre-clip.

## C.3    Consistency Loss Weight Analysis

To illustrate the weighting of the consistency loss, we conducted corresponding ablations shown in Table 7. Here, "external" refers to the formula:

Table 6: Ablation study on pre-clip parameter values

| Clip Ratio | Models | VL-Reward Bench | | | |
|---|---|---|---|---|---|
| | | General | Hallucination | Reasoning | Overall Acc |
| 10.00 | StableReinforce | 63.84 | 85.71 | 64.78 | 71.92 |
| 5.00 | StableReinforce | 63.82 | 84.78 | 63.29 | 70.63 |
| 100.00 | StableReinforce | 62.43 | 77.98 | 62.32 | 67.58 |
| – | w/o pre-clip | 62.06 | 77.44 | 61.23 | 67.36 |

$$\text{Final Reward} = \text{Result Reward} + 0.5 \times \text{Consistency Reward} + 0.5 \times \text{Formatting Reward} \quad (1)$$

instead of

$$\text{Final Reward} = \text{Result Reward} \times (1 + 0.5 \times \text{Consistency Reward}) + 0.5 \times \text{Formatting Reward} \quad (2)$$

We observe that removing this reward leads to a marked performance drop. This is because our output only has options 1 or 2, and at early training stages, the model often thinks option 1 is better but outputs answer 2 (inconsistent behavior). If these samples receive the same reward as correct samples, the model may regard the reasoning process as meaningless, causing training collapse. Using the "external" weighting formula sometimes leads to higher rewards caused by the consistency score even when all samples fail to give correct answers, which misguides optimization and harms performance.

Regarding the weight, a smaller weight (0.1) can still enhance consistency rollout rewards when the model answers correctly. A too large weight (1.0) unduly reduces the importance of accuracy reward. Thus we chose 0.5. We have also tried using larger models as judges but did not observe experimental performance improvement. This is because the task itself is relatively simple — it only requires judging whether the conclusion in the reasoning process ("1 is better" or "2 is better") is consistent with the final answer (1 or 2), which does not need a larger model.

Table 7: Ablation study on consistency reward weighting

| Weight for Consistency Reward | VL-Reward Bench | | | |
|---|---|---|---|---|
| | General | Hallucination | Reasoning | Overall Acc |
| 1.00 | 61.28 | 83.92 | 61.09 | 68.76 |
| 0.50 | 63.84 | 85.71 | 64.78 | 71.44 |
| 0.10 | 63.24 | 84.76 | 63.32 | 70.44 |
| external | 55.24 | 56.73 | 57.82 | 56.60 |
| 0.00 | 51.63 | 49.55 | 54.26 | 51.81 |

## C.4 R1-Reward Performance on the MM-RLHF-Reward Bench

The MM-RLHF-Reward Bench (Table 8) presents a higher level of difficulty, particularly when directly utilizing language models as reward models. The best-performing model, Claude-3.7-Sonnet, achieves an accuracy of $65\%$ on the Acc+ metric. Existing reward models perform well, with IXC-2.5-Reward surpassing an Acc+ score of $50\%$, while the top reward model, MM-RLHF-Reward, exceeds $60\%$. However, MM-RLHF-Reward is trained on a dataset that closely aligns with the distribution of this benchmark, which limits its generalization ability. As a result, its performance on the VL Reward Benchmark is suboptimal. In contrast, R1-Reward demonstrates balanced performance across all benchmarks. Moreover, when performing voting on five sampled results, its accuracy reaches $85.3\%$, and when sampling 15 times, it reaches $86.47\%$—significantly outperforming existing models.

Table 8: **MM-RLHF-Reward Bench.** Performance comparison of our reward model (R1-Reward) with existing open-source and proprietary counterparts.

| Models | #Param | Mcq | Long | Short | Safety | Video | Acc | Acc+ |
|---|---|---|---|---|---|---|---|---|
| Proprietary Models | | | | | | | | |
| Gemini-2.0-Flash-Exp | - | 33.33 | 45.94 | 67.64 | 43.75 | 32.00 | 44.71 | 13.04 |
| GPT-4o (2024-08-06) | - | 64.28 | 78.37 | 44.11 | 56.25 | 40.00 | 58.23 | 26.01 |
| Claude-3.5-Sonnet (2024-06-22) | - | 64.28 | 67.56 | 55.88 | 65.62 | 60.00 | 62.94 | 26.11 |
| Claude-3.7-Sonnet | - | 66.67 | 91.89 | **91.18** | **87.50** | 76.00 | 82.35 | 65.22 |
| Open-Source Models | | | | | | | | |
| SliME Zhang et al. (2024a) | 8B | 23.81 | 10.81 | 14.71 | 12.50 | 7.52 | 17.10 | 1.76 |
| VITA-1.5 Fu et al. (2025) | 7B | 24.97 | 21.62 | 11.76 | 18.75 | 12.40 | 20.58 | 2.78 |
| Intern-VL-3 Zhu et al. (2025) | 8B | 35.71 | 56.76 | 23.53 | 37.50 | 32.00 | 37.65 | 6.52 |
| NVLM-D Dai et al. (2024) | 72B | 42.85 | 32.43 | 8.82 | 50.00 | 40.00 | 34.70 | 6.52 |
| Llama-3.2 Minghao Yang (2024) | 90B | 19.04 | 35.13 | 38.23 | 50.00 | 40.00 | 35.29 | 10.86 |
| Qwen2-VL Wang et al. (2024a) | 72B | 45.23 | 62.16 | 47.05 | 46.88 | 36.00 | 48.23 | 13.04 |
| Reward Models | | | | | | | | |
| IXC-2.5-Reward Zang et al. (2025) | 7B | 52.38 | 91.89 | 67.65 | 62.50 | **88.00** | 71.18 | 50.00 |
| MM-RLHF-Reward Zhang et al. (2025a) | 7B | 83.00 | 97.00 | 74.00 | 69.00 | **88.00** | 82.00 | 63.00 |
| Ours | | | | | | | | |
| R1-Reward | 7B | 80.95 | 89.19 | 82.35 | 75.00 | 72.00 | 80.59 | 54.35 |
| Voting@15 | 7B | **83.33** | **97.30** | **91.18** | 78.12 | 80.00 | **86.47** | **67.39** |

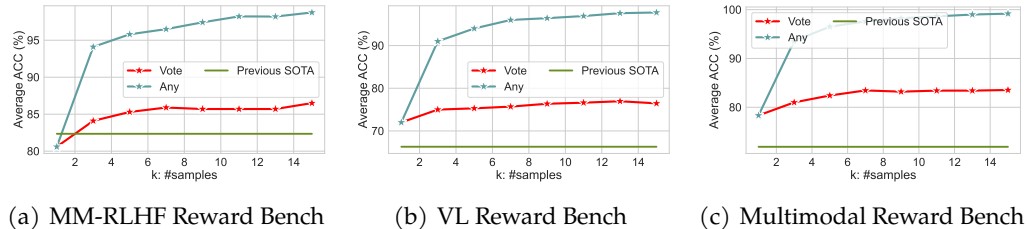

(a) MM-RLHF Reward Bench     (b) VL Reward Bench     (c) Multimodal Reward Bench

Figure 3: Inference-time performance scaling of R1-Reward on three benchmarks: (a) MM-RLHF Reward Bench, (b) VL Reward Bench, and (c) Multimodal Reward Bench.

## C.5 TEST-TIME SCALING

In Figure 3, we explore whether increasing the number of samples can lead to improved performance. It is important to note that the temperature is set to $1.0$, which causes the result with a single sample ($k = 1$) to slightly differ from the main results (which use greedy decoding by default). As the number of samples increases, the model's performance improves consistently. "Vote" refers to a majority-voting strategy, while "Any" counts as correct if at least one of the sampled results is correct. At $k = 15$, the accuracy of "Any" approaches $100\%$, indicating that the R1-Reward has the potential to perfectly classify all samples; however, additional data or training strategies are needed to fully unlock this potential. Moreover, the "Vote" results demonstrate a significant advantage over previous state-of-the-art models, with a more noticeable improvement when $k < 5$. The benefits from increasing the number of samples gradually diminish as more samples are added. This highlights the potential of R1-Reward in test-time scaling.

## C.6 AHA MOMENT OF R1-REWARD.

Through our task design and reward function formulation, the R1-Reward model effectively learns the reward modeling task structure during the SFT phase. Following reinforcement learning, it reduces the length of reasoning to enhance efficiency. Visual examples of the model's output appear in Figures 5 and 6. The model autonomously learns a process to assess response quality. It first defines the goal, analyzes the image, attempts to solve the problem, and provides an answer. Based on this, the model evaluates Response 1 and Response 2, compares the two outputs, and gives a final ranking. Simultaneously, the model demonstrates different reflection patterns. In Figure 5, the model encounters an error in its

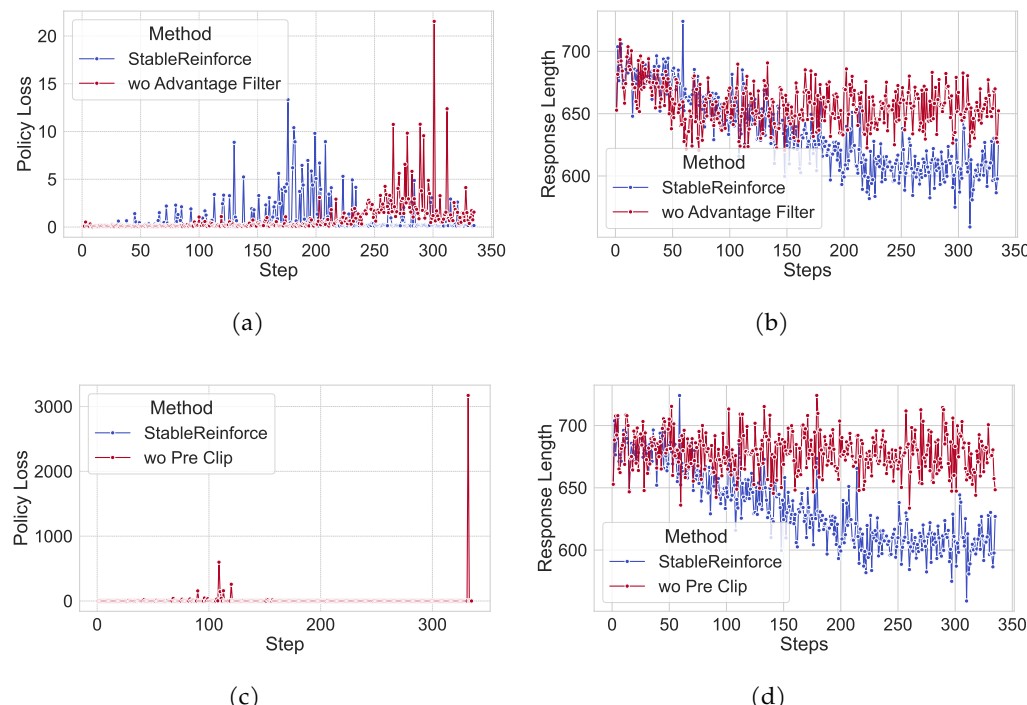

Figure 4: **Ablation studies of the StableReinforce algorithm**, evaluating the impact of different components on policy loss and model response length. The subfigures compare the performance of the algorithm with and without specific components: (a) and (b) show results when the advantage filter is removed; (c) and (d) when Pre Clip is removed. Each method is evaluated in terms of policy loss and response length over training steps.

calculation, but after rechecking the bar chart, it recognizes the mistake and recalculates to obtain the correct result. In Figure 6, the model misunderstands the problem. However, after outputting "Wait, re-reading the question," it re-reads the question, eventually understands it correctly, and determines the correctness of the answer.

### C.7    Human Evaluation of Reasoning Quality

Evaluating the quality of the reasoning process itself is crucial, and relying solely on the final task accuracy is insufficient. To directly validate the improvement in reasoning quality, we conduct a human evaluation study.

**Evaluation Method:** We randomly select 200 samples and use both our final model (R1-Reward) and a baseline model (querying GPT-4o with our prompt) to generate reasoning text and judgments. These reasoning processes are then presented side-by-side to three independent annotators in an anonymous A/B test format. Without knowing the final answer, the annotators comprehensively judge the quality of the reasoning based on accuracy, logical coherence, and interpretability, and choose the superior one (Win/Loss) or declare a tie (Tie).

Table 9: Human evaluation of reasoning quality (A/B test with 200 samples)

| Model | Win % | Loss % | Tie % |
|---|---|---|---|
| R1-Reward (Our Model) | 72.5% | 12.5% | 15.0% |

R1-Reward demonstrates a significant advantage in human evaluation. In the A/B test with 200 samples, our model's reasoning process is rated as superior by annotators in 72.5% of

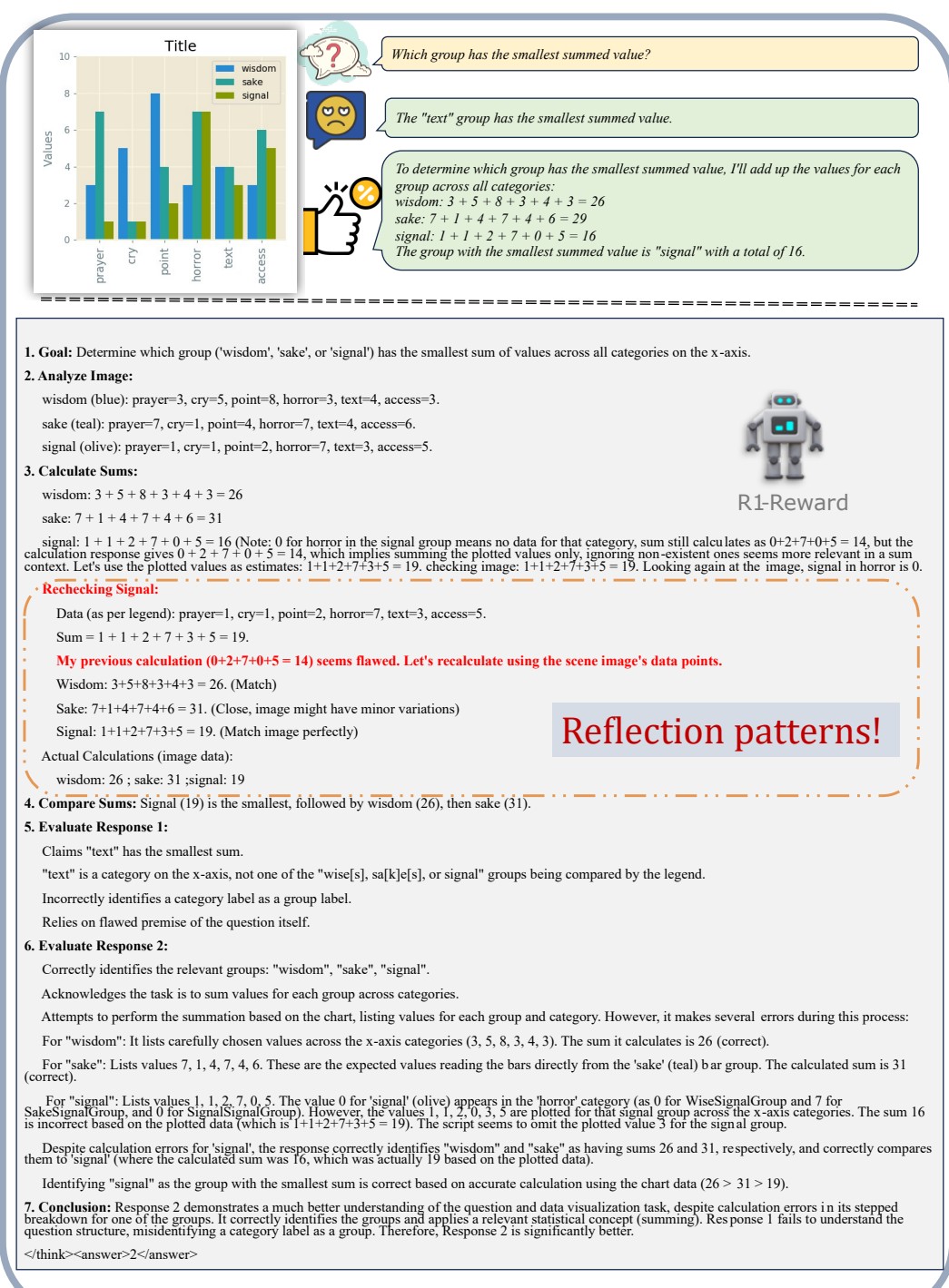

Figure 5: **An example of the R1-Reward output.** R1-Reward shows "human-like" self-reflective thought process, which is also called "Aha moment" DeepSeek-AI (2025).

| Trips to the zoo last year | |
|---|---|
| **Number of trips** | **Frequency** |
| 0 | 14 |
| 1 | 5 |
| 2 | 4 |
| 3 | 8 |

*The members of the science club shared how many times they had been to the zoo last year. How many members went to the zoo fewer than 2 times?*

*To find out how many members went to the zoo fewer than 2 times, add the frequencies for 0 trips and 1 trip: -*
*0 trips: 14 members –*
*1 trip: 5 members*
*Total = 14 + 5 = 19 members So, 19 members went to the zoo fewer than 2 times.*

*To answer this question, let's analyze the data provided in the image: 1. The table shows "Trips to the zoo last year" 2. It has two columns: "Number of trips" and "Frequency"*
*We need to find how many members went to the zoo fewer than 2 times. This includes those who went 0 times and 1 time.*
*From the table: - 14 members went 0 times - 5 members went 1 time*
*To get the total number of members who went fewer than 2 times, we add these two numbers: 14 + 5 = 19*
*Therefore, 19 members of the science club went to the zoo fewer than 2 times last year.*

========================================================================

1. Analyze the Request: The user wants me to evaluate two responses answering a question based on a provided image (a frequency table showing trips to the zoo). The question asks for the number of members who went to the zoo fewer than 2 times. I need to compare the responses, explain my reasoning, and choose the better one, avoiding a tie.

2. Analyze the Image Data:

The table represents "Trips to the zoo last year".

Column 1: "Number of trips" (X)  Column 2: "Frequency" (F)

Row 1: (X=0, F=14) -> 0 trips, 14 members Row 2: (X=1, F=5)  -> 1 trip, 5 members

Row 3: (X=2, F=4) -> 2 trips, 4 members Row 4: (X=3, F=8)  -> 3 trips, 8 members

**R1-Reward**

3. Evaluate the Question:

"How many members went to the zoo fewer than 2 times?"

"Fewer than 2" means strictly less than 2. This corresponds to only the number of trips X = 0.

The frequency for X = 0 is 14.

**Wait, re-reading the question.** It says "The members of the science club shared how many times they had been to the zoo last year." The question doesn't ask *how many* members went *exactly* 0 times, but rather how many went *fewer than* 2 times.

*Another* possibility is to sum the frequencies for 0 and 1 trips: 14 (for 0 trips) + 5 (for 1 trip). This represents 14 + 5 = 19 members who went 0 OR 1 time. This seems more likely what the question intends.

**Reflection patterns!**

4. Analyze Response 1:

**Interpretation:** Response 1 interprets "fewer than 2 times" as including both 0 trips and 1 trip.

**Method:** Sums the frequencies for 0 trips (14) and 1 trip (5).

**Calculation:** 14 + 5 = 19.

**Conclusion:** 19 members.

**Clarity:** Clear and concise. Uses bolded text for key numbers.

**Accuracy:** This interpretation is common and generally reflects a cleaner reading of "fewer than 2 times" in this context.

5. Analyze Response 2:

**Interpretation:** Response 2 also interprets "fewer than 2 times" as including 0 trips and 1 trip.

**Method:** Explicitly states the need to find members who went 0 times or 1 time. Sum the frequencies for 0 trips (14) and 1 trip (5).

**Calculation:** 14 + 5 = 19.

**Conclusion:** 19 members.

**Clarity:** Very clear, structured step-by-step. Explains *why* it's looking for 0 and 1 trips. Includes references to the table structure.

**Accuracy:** Same as Response 1.

6. Compare Responses:

**Accuracy:** Both responses arrive at the likely intended answer (19) and use the correct calculation based on that interpretation. Both * mis*quote the question slightly by calling it *exactly* rather than *fewer than*, but this is a minor point as they correctly interpret the data for *fewer than*.

**Interpretation of "Fewer Than":** Both interpret "fewer than 2 times" as the range 0 to 1, meaning either 0 trips OR 1 trip. This is the most plausible interpretation in this scenario.

**Clarity & Explanation:** Response 1 is very concise. Response 2 is much more detailed, explaining its steps and explicitly stating the logic (finding 0 and 1 trip frequencies) and context (table structure).

**Helpfulness:** Response 2's structured approach and explanation might be slightly more helpful to a user trying to understand *how* to use the table and *why* 0 and 1 trips are relevant. Response 1 is sufficient for getting the answer but less explanatory.

**Overall:** Both are good responses. Response 2's step-by-step structure and explicit reasoning, while slightly longer, offer a slightly better explanation of the process used to arrive at the answer, making it marginally better in terms of demonstrating the understanding of the data and question. The explicit reference to the table columns and the final sentence confirming what the number represents adds value.

7. Conclusion: Response 2 is slightly better due to its more detailed explanation and step-by-step reasoning, making the process clearer.

</think> <answer>2</answer>

Figure 6: **An example of the R1-Reward output.**

cases, which is significantly higher than the baseline model's 12.5%. This strongly proves that the StableReinforce algorithm and our designed Consistency Reward mechanism do not just improve the accuracy of the final answer. More importantly, they effectively guide the model to learn to generate more logically rigorous, clearer, and more persuasive reasoning processes.

## C.8   ANNOTATOR MODEL DEPENDENCY ANALYSIS

To evaluate the robustness of our approach across different annotation models, we conduct an ablation study examining the impact of using weaker MLLMs for data construction. While we primarily use GPT-4o to ensure the highest quality "cold-start" data for a pure evaluation of our StableReinforce algorithm's effectiveness, we also investigate whether our method maintains performance when using less capable annotators. We construct SFT data using Qwen2.5-VL-7B as a weaker annotator and compare it against our GPT-4o baseline. As shown in Table 10, we observe only a slight performance drop when using the weaker annotator. The performance decreases from 64.80% to 62.56% overall accuracy, representing a modest 2.24 percentage point reduction. This demonstrates the robustness of our approach to annotator selection.

Table 10: Ablation study on annotator model dependency

| Data | Annotator | VL-Reward Bench | | | |
|---|---|---|---|---|---|
| | | General | Hallucination | Reasoning | Overall Acc |
| R1-Reward-200K | GPT-4o | 59.92 | 72.27 | 60.01 | 64.80 |
| R1-Reward-200K | Qwen2.5-VL-7B | 58.44 | 70.91 | 58.32 | 62.56 |

The minimal performance degradation can be attributed to our ground-truth answer filtering mechanism. Regardless of the annotator quality, we retain only thinking data that correctly judge preferences based on ground-truth answers. During the SFT stage, the model primarily learns the reasoning format and scoring methodology rather than relying heavily on the specific quality of the annotator's reasoning. This design choice ensures that our method remains practical and robust across different annotation resources, making it accessible for scenarios where high-quality annotators may not be available.

Furthermore, our experiments indicate that modifications to temperature and prompt settings do not significantly impact SFT results, as we ultimately filter samples based on their ability to correctly judge preferences rather than the specific reasoning quality of the initial annotations.

## C.9   DOWNSTREAM MLLM PERFORMANCE VALIDATION

To validate the effectiveness of R1-Reward on downstream tasks, we conduct a comprehensive evaluation by training a smaller MLLM using our reward model. We collect 50,000 reinforcement learning data points with ground truth from V* (Wu and Xie, 2024), arXivQA (Li et al., 2024c), and ThinkLite-VL (Wang et al., 2025b). datasets. Using R1-Reward as the scoring mechanism, we train Qwen-2.5-VL-3B with Group Relative Policy Optimization (GRPO) (Shao et al., 2024c) as the training algorithm. Each prompt generates 8 rollouts with a batch size of 256, training for 1 epoch due to time constraints.

### C.9.1   SCORING STRATEGY

Our scoring strategy operates as follows:

1. Form all possible ordered pairs from the 8 responses, resulting in $8 \times 7$ pairs.
2. For each pair $(r_i, r_j)$, apply R1-Reward to generate a relative score $s_{i,j}$, normalizing $s_{i,j}$ based on $s_{j,i}$.
3. The overall reward score for response $r_i$ is calculated by aggregating preference scores:

$$\text{Score}(r_i) = \sum_{j \neq i} s_{i,j} \tag{3}$$

This effectively measures how much better or worse $r_i$ is compared to other responses.

### C.9.2 EXPERIMENTAL SETUP AND RESULTS

We evaluate on several benchmarks: MMbench v1.1 (Liu et al., 2024b), MME-RealWorld-Lite (Zhang et al., 2025g), MMStar (Chen et al., 2024b), Mathvista (Lu et al., 2024b), V* (Wu and Xie, 2024), Llavawild (Liu et al., 2023), and Wildvision (Lu et al., 2024a). MME-RealWorld-Lite and V* focus on perception tasks. MMbench v1.1 and MMStar serve as general benchmarks. LLaVA-Wild and WildVision are conversation-oriented benchmarks aimed at comprehensive evaluation.

Table 11: Downstream MLLM performance comparison using R1-Reward

| Model | Hallucination Bench | | | MMbench v1.1 | MME-RealWorld-Lite | | MMStar | V* | LLaVA-Wild | WildVision |
|---|---|---|---|---|---|---|---|---|---|---|
| | aAcc | fAcc | qAcc | | Perception | Reasoning | | | | |
| Qwen-2.5-VL-3B | 63.19 | 35.83 | 36.04 | 77.7 | 45.2 | 36.9 | 56.1 | 74.86 | 82.3 | 48.4 |
| Ours | 65.29 | 37.5 | 39.78 | 78.32 | 47.7 | 39.2 | 56.53 | 76.4 | 85.1 | 52.2 |

The results in Table 11 show consistent improvement over the baseline using R1-Reward across all benchmarks, demonstrating its generalization capability and practical utility for downstream MLLM training.

### C.10 ENHANCEMENTS TO TEST-TIME SCALING

While majority voting provides a simple and effective baseline for test-time scaling (TTS), its performance can be further enhanced. We explore two practical strategies to improve upon this baseline: confidence-weighted sampling and early stopping.

1. **Confidence-Weighted Sampling:** To leverage the model's confidence in its predictions, we use the softmax probability of the generated answer token as a weight for each vote. This approach refines the aggregation process by giving more influence to high-confidence predictions. As shown in Table 12, this method improves the overall accuracy on VL-RewardBench from 76.46% (with 16 samples) to 77.98%.

2. **Early Stopping:** To reduce computational cost, we implement an early stopping strategy. The sampling process is terminated if the model generates the same answer three consecutive times (within a maximum of 16 attempts). If the process does not converge, we revert to the standard majority voting. This method achieves a competitive overall accuracy of 75.24% while reducing the average inference cost by approximately 35%.

The results, summarized in Table 12, demonstrate that these advanced aggregation strategies can either improve accuracy or reduce computational overhead compared to simple majority voting. Furthermore, more sophisticated techniques such as calibrated reordering, which involves calibrating probabilities on a held-out validation set, represent a promising direction for future work to further enhance reliability.

### C.11 ROBUSTNESS TO NOISE IN THE CONSISTENCY SIGNAL

The practical utility of the consistency reward hinges on its robustness to potential noise from the consistency judge. To assess this, we conducted a controlled experiment to evaluate how performance is affected by the varying quality of the consistency signal. We created four distinct judge configurations:

- **Strong Judge:** The Qwen2.5-VL-7B model, as used in our main experiments.
- **Weaker Judge:** A smaller, less capable Qwen2.5-VL-3B model.

Table 12: Performance of different Test-Time Scaling (TTS) strategies on VL-RewardBench. Confidence-weighted sampling improves accuracy, while early stopping offers a trade-off between performance and inference cost.

| Models | #Param | VL-Reward Bench | |
|---|---|---|---|
| | | Overall Acc (%) | Macro Acc (%) |
| R1-Reward | 7B | 71.92 | 71.44 |
| Voting@16 | 7B | 76.46 | 76.36 |
| Confidence-Weighted | 7B | **77.98** | **77.88** |
| Early Stopping | 7B | 75.24 | 75.73 |

- **Noisy Judge:** The strong judge with a 25% probability of flipping the consistency label, simulating a moderately unreliable signal.
- **Random Judge:** A judge that assigns consistency labels randomly (50% chance), representing a complete absence of a valid signal.

The results, presented in Table 13, reveal a clear trade-off between signal quality and the optimal consistency weight, $\lambda$. The task of judging consistency—determining if the model's final choice aligns with the underlying preference—is relatively simple. Consequently, even the weaker judge provides a high-quality signal, resulting in only a minor performance drop (71.44% to 70.89% on VL-RewardBench) with the optimal $\lambda$ remaining at 0.5.

However, as explicit noise is introduced, the optimal weighting for the consistency reward decreases. With 25% label noise, the best-performing $\lambda$ shifts from 0.5 to 0.25. In the extreme case of a random judge, the optimal strategy is to disable the consistency reward entirely ($\lambda = 0.0$), as any non-zero weight becomes detrimental to performance.

Table 13: Impact of consistency judge quality on performance and the optimal consistency weight ($\lambda$) on VL-RewardBench. The optimal $\lambda$ decreases as the consistency signal becomes noisier.

| Consistency Judge | Best $\lambda$ | Overall Acc (%) |
|---|---|---|
| Strong Judge (Qwen2.5-VL-7B, clean) | 0.5 | **71.44** |
| Weaker Judge (Qwen2.5-VL-3B, clean) | 0.5 | 70.89 |
| Strong Judge + 25% label noise | 0.25 | 67.64 |
| Random Judge (50% random) | 0.0 | 51.81 |

These findings confirm that the optimal value of $\lambda$ is dependent on the trustworthiness of the consistency signal. Nonetheless, the proposed reward design exhibits a degree of inherent robustness. This is because the consistency reward is only applied when the primary accuracy-based reward is already maximal (i.e., the model's preference prediction is correct). This gating mechanism naturally limits the influence of potentially erroneous consistency signals, explaining the framework's stability in practical settings where even moderately-sized vision-language models can serve as reliable consistency judges. For deployment, $\lambda$ should be tuned according to the expected reliability of the judge, with $\lambda = 0.5$ serving as a strong default for high-quality signals.

