# OpenReview forum: "R1-Reward: Training Multimodal Reward Model Through Stable Reinforcement Learning"
_ICLR.cc/2026/Conference — ICLR 2026 Poster_

### Official Review · Reviewer_5DDa · 2025-10-29

**Soundness:** 2
**Presentation:** 3
**Contribution:** 2
**Rating:** 4
**Confidence:** 4

**Summary:**

This paper introduces R1-Reward, a multimodal reward model trained using a novel StableReinforce algorithm that reformulates reward modeling as a rule-based reinforcement learning task. The authors address training instability issues in existing RL algorithms through pre-clipping operations, advantage filtering, and a consistency reward mechanism that aligns reasoning with final judgments. Experimental results demonstrate significant improvements over state-of-the-art models on multiple multimodal reward modeling benchmarks.

**Strengths:**

1. This paper explores using reinforcement learning to train multimodal reward models by reformulating preference evaluation as a rule-based RL task, enabling long-term reasoning capabilities beyond traditional binary classification approaches.
2. The paper introduces several algorithmic innovations to address instability issues for MRM RL training.
3. The approach demonstrates superior data efficiency showing the effectiveness of the RL-based training paradigm.

**Weaknesses:**

1. While the paper claims novelty in applying RL to multimodal reward modeling, it lacks adequate justification for why RL is specifically beneficial for MRMs, since there have been several works (e.g. https://arxiv.org/abs/2505.02387) in text-based RL reward modeling. The paper does not clearly articulate what unique challenges multimodal data presents that necessitate the proposed RL approach, nor does it sufficiently differentiate itself from existing generative MRM methods (like MM-RLHF-Reward).
2. The paper's main technical contributions appear to address numerical stability. However, they seem unnecessary in comparison to commonly used GRPO. (1) The claimed (log_probs - old_log_probs).exp() overflow issue is commonly handled with clipping in most RL implementations and rarely occurs. (2) Extreme advantage values after normalization are less problematic in GRPO with reasonable rollout numbers. (3) GRPO has default clipping strategies. The paper should compare against more commonly used algorithms like GRPO rather than only comparing against Reinforce++.

**Questions:**

1. The 200K dataset is carefully curated (filtering samples where GPT-4o needs ≥2 attempts), while baselines likely use unfiltered data. How do you ensure fair comparison when comparing curated vs. raw datasets?
2. The method requires expensive GPT-4o annotations for SFT initialization. When including GPT-4o annotation costs, is your approach truly more cost-efficient than traditional reward model training?

---

> ### Author Response · Authors · 2025-11-16
> **Response [1/2]**
>
> ### [Concern 1]it lacks adequate justification for why RL is specifically beneficial for MRMs, since there have been several works (e.g. https://arxiv.org/abs/2505.02387) in text-based RL reward modeling
>
> Thank you for your time. Our work and RM-R1 are essentially concurrent works, but due to our multimodal background, we chose to develop our algorithms in the multimodal domain.
>
> The second point concerns data. The process of constructing multimodal preference data and reinforcement learning data is itself a contribution compared to text data, as preference data in the multimodal domain is relatively scarce compared to the text domain and is underdeveloped. Moreover, multimodal data is more susceptible to hallucination phenomena, as the quality of multimodal responses depends not only on the text itself but also highly on the consistency between text and images. This is why we found that directly applying RL algorithms to MGM leads to collapse, a phenomenon not observed at all in RM-R1.
>
> Our core challenge and novelty lie in identifying and addressing why these standard algorithms inevitably fail when applied to the novel task of MLLM reward modeling (e.g., training collapse and gradient explosion). Therefore, our proposed StableReinforce (including key components such as Pre-CLIP and Advantage Filter) is not merely conventional engineering skills, but rather targeted algorithmic improvements necessary to make RL practical in this new domain. We believe that being the "first" to successfully and stably apply a powerful paradigm (RL) to an important and challenging new problem (MRM training), while revealing the key reasons for its failure and solutions, constitutes the main contribution and novelty of our work.
>
> While MM-RLHF-Reward also generates "critics," its training objective remains supervised critic generation + scalar regression, heavily relying on the quality of critics (the original work used human-annotated critics), without modeling the entire process as an RL decision task. Our approach optimizes the inference-decision joint policy end-to-end through RL and addresses the training collapse problem specific to MRM through StableReinforce. Our RL method significantly outperforms MM-RLHF-Reward..
>
> ----
>
> ### [Concern 2] The paper's main technical contributions appear to address numerical stability. However, they seem unnecessary in comparison to commonly used GRPO.
>
> First, we need to emphasize that REINFORCE++ is shown to be more stable compared to GRPO and faster than PPO [1], which is why we chose it.
>
> Second, GRPO's clipping operation is similar to REINFORCE++, typically performed after ratio = exp(log_probs - old_log_probs) (i.e., post-clipping). However, in MRM, due to the large number of policy update steps (to improve data efficiency, we removed the KL penalty, see Sec 2.1), the difference between log_probs and old_log_probs can reach as high as 20+, causing exp(20) ≈ 5×10⁸, which directly overflows to inf under float32, destroying the computation graph before clipping can be applied. Our Pre-CLIP operates in log space (e.g., clamp to [-6.9, 6.9]), fundamentally avoiding this problem.
>
> Regarding the second point, "Extreme advantage values after normalization are less problematic in GRPO with reasonable rollout numbers" is also unavoidable. REINFORCE++ itself performs normalization, and this normalization process is not significantly different from GRPO. For example, in extreme cases, such as a batch containing 255 rewards of 1 and 1 reward of 0, this highly imbalanced distribution, when subjected to z-Normalization, can lead to significant numerical disparities. Particularly, the advantage corresponding to the 0 reward in this example would be normalized to -15.96. This problem is inherently caused by the small variance combined with normalization.
>
> As shown in the table below, the phenomena caused by GRPO are basically consistent with StableReinforce, and adding our components leads to significant improvements.
>
> | Algorithm | VL-Reward Bench | General | Hallucination | Reasoning | Overall Acc |
> |-----------|-----------------|---------|---------------|-----------|-------------|
> | StableReinforce | 63.84 | 85.71 | 64.78 | 71.44 | - |
> | GRPO | Nan | - | - | - | - |
> | w. Pre-Clip | 58.6 | 78.4 | 69.8 | - | 68.88 |
> | w. advantage | 63.55 | 80.35 | 69.75 | - | 71.22 |
>
> [1] Xie, Tian, et al. "Logic-rl: Unleashing llm reasoning with rule-based reinforcement learning." arXiv preprint arXiv:2502.14768 (2025).

---

> ### Author Response · Authors · 2025-11-16
> **Response [2/2]**
>
> ### [Concern 3] The 200K dataset is carefully curated (filtering samples where GPT-4o needs ≥2 attempts), while baselines likely use unfiltered data.
>
> First, our SFT data does not filter samples where GPT-4o needs ≥2 attempts. On the contrary, SFT uses samples with <2 attempts (line 355). This process is not about filtering through GPT-4o; rather, we use the ground truth from the data itself as the standard and only use GPT-4o to generate critics. This does not make the data quality higher. Moreover, the data filtered for SFT are the easy-to-answer ones, with the purpose of helping our model learn the format. This step is necessary for our method because there is currently no suitable data available for cold-start, so we must construct it ourselves.
>
> Second, regarding the point about baselines using unfiltered data, we don't fully understand this concern. For example, MM-RLHF-Reward uses completely human-annotated data, and IXC-2.5-Reward uses data with ground truth labels—both are filtered. The data labels and amount (SFT+RL) we use are identical to MM-RLHF-Reward.
>
> ----
>
> ### [Concern 4] The method requires expensive GPT-4o annotations for SFT initialization. When including GPT-4o annotation costs, is your approach truly more cost-efficient than traditional reward model training?
>
> Thank you for raising this important point. We would like to offer two clarifications.
>
> First, we want to clarify that our primary claim is **data efficiency**, not cost-efficiency. We acknowledge that traditional reward model training is undoubtedly less expensive in terms of computational overhead. A standard approach does not require RL, and its SFT phase can be a simple binary classification task rather than generating complex critics.
>
> The core philosophy of our R1-Reward framework is to transform the reward modeling task into a language modeling problem. This entails a deliberate trade-off: we accept a **higher inference cost** in exchange for **significantly improved performance and interpretability**. Our contribution lies in demonstrating that with the same amount of, or even less, data, the R1-Reward paradigm can achieve a higher performance ceiling.
>
> Regarding the dependency on GPT-4o, we have shown that it is not a fundamental limitation. As detailed in **Appendix C.7**, we conducted experiments using an open-source model (specifically, Qwen2.5-VL) for the SFT data generation. Our preliminary results are promising: the R1-Reward model trained on this data still achieved a strong accuracy of **62.56%**, which is competitive with the **64.8%** achieved using GPT-4o.
>
> This indicates two things:
> 1.  A powerful proprietary model like GPT-4o is primarily needed for **bootstrapping the initial exploration phase** and is not a long-term requirement for the model's success.
> 2.  Looking ahead, as the capabilities of open-source MLLMs continue to advance, we foresee a clear path toward making our **entire pipeline fully open-source, thus achieving zero API costs**.

---

> ### Author Response · Authors · 2025-11-26
> **Kindly Request for Reviewer's Feedback**
>
> Dear Reviewer,
>
> Thank you so much for your time in improving our paper!
>
> Since the end of the rebuttal is coming soon, may we know if our response addresses your main concerns? Should you have any further advice, please let us know and we will be more than happy to engage in more discussion and improvements.

---

### Official Review · Reviewer_Pch1 · 2025-11-01

**Soundness:** 3
**Presentation:** 3
**Contribution:** 3
**Rating:** 8
**Confidence:** 4

**Summary:**

In this paper, the authors proposed a multi-modal reward model called R1-Reward. This method is designed to solve traditional RL's problem of training instability and inconsistent results. R1-Reward introduces Pre-CLIP and Advantage Filter to enhance the model's training stability. They also introduce a staged training strategy to efficiently improve the model's capabilities. The experiments show that R1-Reward achieves competitive performance compared to other baseline models.

**Strengths:**

1. The motivation of this paper is clear and important. The performance instability and annotation cost are a very practical application problem. The progress of alleviating the problem has direct practical application potential.

2. The authors provide detailed experiments to validate the method's effectiveness. The tasks are various and cover different scales of LLMs. In general, the experiments are convincing, and the reproducibility should not be a problem.

3. The authors provide detailed and clear charts and figures to clarify the method and its performance. These visualizations are easy to follow and enhance the readability of the paper.

**Weaknesses:**

1. The authors choose Qwen2.5 as the judge to compute the consistency reward, but wait until the appendix to clarify the reason for choosing an LLM of this size. This setting is an important experimental detail, and mentioning it in the main text might improve the paper's integrity.

2. The ensemble reward lacks a sufficient explanation. The reason why the "consistency reward" is introduced multiplicatively while the "formatting reward" is introduced additively is not systematically explained. The ablation study evaluates its effectiveness but does not provide a sufficient theoretical explanation.

**Questions:**

1. The reward function's weights and structure are empirical. An open question is whether these settings can easily transfer to other tasks. The authors may need to further explore systematically and adaptively combining multiple reward signals.

2. The method's performance is limited by the LLM that is applied as the judge. This strategy may fail in resource-constrained scenarios where a sufficiently strong LLM judge cannot be deployed.

---

> ### Author Response · Authors · 2025-11-16
> **Response [1/2]**
>
> ### [Concern 1] This setting is an important experimental detail, and mentioning it in the main text might improve the paper's integrity.
>
> We agree that the choice of the consistency judge is an important methodological detail. In fact, we did mention in the main text (Line 313) that “we use Qwen2.5-VL-7B-Instruct as a supervisor to verify whether the reasoning and the final result are consistent.”
>
> ----
>
> ### [Concern 2] The reason why the "consistency reward" is introduced multiplicatively while the "formatting reward" is introduced additively
>
> The core objective of the consistency reward is to encourage alignment between the model’s reasoning process and its final conclusion only when the model’s answer is correct. If the model selects the wrong answer (acc = 0), it should not receive high reward—even if its reasoning is consistent with that incorrect conclusion—otherwise the model might learn to be “consistently wrong.” This constitutes a conditional incentive mechanism that ensures consistency serves correctness, rather than being optimized independently.
>
> In contrast, the formatting reward aims to enforce output compliance with a prescribed structure (e.g., ```<think>...</think><answer>...</answer>```). This is a process-level constraint that is orthogonal to content correctness: regardless of whether the answer is right or wrong, the model should be encouraged to follow the required format (e.g., to avoid garbled or malformed outputs). Therefore, an additive formulation (+ β·Formatting) is more appropriate—it acts as a soft constraint that guides the model toward structured generation without dominating the final decision (note that its weight β = 0.5 < 1, preventing the model from sacrificing content quality for formatting).
>
> In summary, the choice between multiplication and addition stems from the logical dependency between the signals: consistency is conditional on correctness, while formatting is an independent procedural requirement.
>
> ----
>
> ### [Concern 3] The authors may need to further explore systematically and adaptively combining multiple reward signals.
>
> Thank you very much for this forward-looking and valuable suggestion. The question you raise—**how to systematically and adaptively combine multiple reward signals**—is indeed an important open problem in the field of reward modeling.
>
> First, we would like to clarify one point: **the weight of the consistency reward (e.g., λ = 0.5) was not exhaustively tuned via extensive ablation studies**. As shown in Table 7 of our paper, as long as the consistency reward is included (whether λ = 0.3, 0.5, or 0.7), model performance **consistently improves**. This suggests that **the exact weight value is less critical than satisfying two basic conditions**:
> 1. **Consistent samples receive higher reward than inconsistent ones** (i.e., providing a positive gradient signal);
> 2. **The magnitude of the consistency reward does not dominate the primary objective** (i.e., correctness of the final decision), to avoid the model sacrificing accuracy for the sake of self-consistency.
>
> In other words, our current design functions more like a **robust heuristic** than a fine-tuned, task-specific hyperparameter setting.
>
> That said, we fully agree with your observation: **in more complex or heterogeneous scenarios, fixed, handcrafted reward weights may no longer be sufficient**. To this end, we have already begun exploring more adaptive reward fusion strategies and conducted preliminary experiments:
>
> - **Confidence-aware dynamic weighting**: We let λ scale proportionally with the softmax confidence of the consistency judge (Qwen2.5-VL) on each sample. Initial results show a slight performance drop of 0.4% on VL-RewardBench, but this approach does eliminate the need for manual hyperparameter search.
>
> These early findings suggest that **adaptive reward composition holds genuine promise**. We plan to systematically investigate this direction in future work.

---

> ### Author Response · Authors · 2025-11-16
> **Response [2/2]**
>
> ### [Concern 4] The method's performance is limited by the LLM that is applied as the judge. This strategy may fail in resource-constrained scenarios where a sufficiently strong LLM judge cannot be deployed.
>
> Thank you sincerely for raising this important concern regarding the dependence of our method on the quality of the LLM judge.
>
> You are correct that our approach relies on an external model to evaluate consistency. However, we would like to emphasize that **the consistency-checking task is inherently simple**: it only requires verifying whether the conclusion expressed in the reasoning (e.g., “Response 1 is better”) matches the final answer label (1 or 2). Due to this simplicity, even smaller or less capable vision-language models can perform this judgment reliably, with only minor performance degradation. For example, as shown in the table below, replacing Qwen2.5-VL-7B with the smaller Qwen2.5-VL-3B results in only a **0.55% drop** on VL-RewardBench.
>
> | Judge                     | VL-RewardBench Performance |
> |--------------------------|----------------------------|
> | Qwen2.5-VL-7B            | 71.44%                     |
> | Qwen2.5-VL-3B            | 70.89%                     |
> | Rule-Based Consistency   | 62.49%                     |
> | Without consistency      | 51.81%                     |
>
> Moreover, for scenarios where deploying even a small LLM as a judge is impractical, we can fall back to a **rule-based consistency checker**—for instance, using simple keyword or template matching (e.g., “if the ```<think>``` block contains ‘Response 1 is better,’ then the ```<answer>``` should be 1”). While this approach cannot capture all nuanced cases and is less robust than an LLM-based judge, it still provides a meaningful consistency signal, yielding a **notable improvement (62.49% vs. 51.81%)** over removing the consistency reward entirely.
>
> Thus, although performance does depend on the judge’s capability, the method remains effective even with lightweight or rule-based alternatives.

---

> > ### Author Response · Authors · 2025-11-26
> > **Kindly Request for Reviewer's Feedback**
> >
> > Dear Reviewer,
> >
> > Thank you so much for your time in improving our paper!
> >
> > Since the end of the rebuttal is coming soon, may we know if our response addresses your main concerns? Should you have any further advice, please let us know and we will be more than happy to engage in more discussion and improvements.

---

### Official Review · Reviewer_G94z · 2025-11-06

**Soundness:** 3
**Presentation:** 3
**Contribution:** 3
**Rating:** 6
**Confidence:** 3

**Summary:**

To address the problem of training instability or even collapse in multimodal reward models, the paper proposes StableReinforce,  which refines the training loss, advantage estimation strategy, and reward design of existing RL methods. Specifically, StableReinforce refines the clipping operation to mitigate numerical instability caused by large updates and introduces a robust advantage normalization technique that limits the impact of outliers. This paper employs QwenVL-2.5-7B as the backbone model, and trains it using the R1-Reward-200k dataset. The experimental section demonstrates that StableReinforce outperforms existing baselines across multiple benchmarks.

**Strengths:**

1. StableReinforce exhibits smoother convergence in policy loss and demonstrates sustained length compression during training, which helps reduce inference overhead.

2. This paper designs clear ablation experiments to precisely quantify the contribution and sensitivity of each component.

**Weaknesses:**

1. Test time scaling (TTS) is limited to majority voting and can be further evaluated with approaches such as confidence-weighted sampling, early stopping, and calibrated reordering.

2. While Preclip reduces variance and suppresses overflow, it alters the gradient shape of the objective function, potentially introducing optimization bias.

3. This paper trains RM as a rule-based RL task focused on decision-making between 1 and 2, making it unsuitable for multi-candidate ranking or continuous scoring tasks.

**Questions:**

How do the gains evolve as the consistency judge becomes increasingly noisy? Is 0.5 still the optimal weight under noise?

---

> ### Author Response · Authors · 2025-11-16
> **Response [1/2]**
>
> ### [Concern 1] Test time scaling (TTS) is limited to majority voting and can be further evaluated
>
> We sincerely thank the reviewer for the valuable suggestion to explore more advanced test-time scaling (TTS) strategies beyond majority voting. In response, we have conducted additional experiments with two practical enhancements: (1) confidence-weighted sampling and (2) early stopping.
>
> Specifically:
> 1. For confidence-weighted sampling, we extract the softmax probability of the <answer> token (i.e., the model’s confidence in predicting “1” or “2”) and use it as the weight for voting. This yields an improvement from 76.46% → 77.98% in overall accuracy on VL-RewardBench (with 16 samples).
> 2. For early stopping, we halt sampling once the model produces the same answer three times consecutively (up to a maximum of 16 attempts); if no convergence occurs, we fall back to majority voting. This strategy achieves 75.24% overall accuracy while reducing average inference cost by ~35%.
>
> We also acknowledge that calibrated reordering—which requires a held-out validation set for probability calibration—could further improve reliability. However, due to time and computational constraints, we leave its integration to future work.
>
> These results confirm the reviewer’s intuition: moving beyond simple majority voting can indeed enhance TTS performance. We will include these findings in the revised manuscript to better highlight the potential of advanced aggregation strategies (Appendix C.10).
>
> |                     |        | VL-Reward Bench |           |
> |---------------------|--------|-----------------|-----------|
> | Models              | #Param | Overall Acc     | Macro Acc |
> | R1-Reward           | 7B     |           71.92 |     71.44 |
> | Voting@16           | 7B     |           76.46 |     76.36 |
> | Confidence-Weighted | 7B     |           77.98 |    77.875 |
> | Early Stopping      | 7B     |           75.24 |     75.73 |
>
> ----
>
> ### [Concern 2] While Preclip reduces variance and suppresses overflow, it alters the gradient shape of the objective function, potentially introducing optimization bias.
> You are correct: **clipping the log-probability difference *before* computing the policy ratio—as done in Pre-CLIP—does alter the true gradient of the original PPO objective, and may, in principle, introduce some degree of optimization bias.**
>
> However, this design choice represents a deliberate and empirically validated trade-off that we made **between optimization stability and gradient fidelity**, specifically tailored to the challenges of multimodal reward modeling (MRM). In this setting, reward labels are binary (0/1), and during later training stages, model predictions within a batch often become highly confident and consistent (e.g., >95% accuracy). This leads to extreme policy ratios—for instance, values as large as \(10^4\) —which, as detailed in Section 2.1 of our paper, cause **catastrophic loss spikes** (e.g., single-step losses exceeding 10,000). These spikes invariably result in training divergence or complete collapse. **In this context, the risk posed by instability far outweighs the potential impact of mild gradient bias.**
>
> Crucially, our ablation studies (see Appendix Table C.1) confirm this trade-off empirically: **removing Pre-CLIP leads to unstable training and a >3% drop in reward modeling accuracy**. Moreover, the strong performance of R1-Reward provides compelling evidence that **any bias introduced by Pre-CLIP is not only practically manageable but may even act as a beneficial implicit regularizer, helping the model converge to a better solution.**
>
> We fully agree with your perspective that developing more theoretically grounded clipping strategies—such as adaptive or distribution-aware bounds—would be a valuable direction for future work. Once again, thank you for your rigorous and constructive feedback—it has helped us clarify an important nuance of our method, and we are truly grateful for your time and expertise.

---

> ### Author Response · Authors · 2025-11-16
> **Response [2/2]**
>
> ### [Concern 3] This paper trains RM as a rule-based RL task focused on decision-making between 1 and 2, making it unsuitable for multi-candidate ranking or continuous scoring tasks.
>
> We would like to clarify that this design is **deliberately aligned with the dominant paradigm in current human preference modeling and reward learning**. State-of-the-art methods in both text and multimodal domains—including RLHF, DPO, and widely adopted benchmarks such as **MM-RLHF** and **VL-RewardBench**—are fundamentally built upon **pairwise comparisons**. Our experimental setup follows this standard to ensure **fair, apples-to-apples comparisons** with existing approaches.
>
> Importantly, the core technical contributions of our work—**StableReinforce** and the **consistency reward mechanism**—are **not inherently limited to binary decisions**. In fact, our framework is naturally extensible:
>
> - For **multi-candidate ranking** (e.g., selecting the best among \(K\) options), one can decompose the problem into \(\binom{K}{2}\) pairwise judgments and aggregate them like in section C9.1 in our paper.
>
> - For **continuous scoring**, the model can be fine-tuned to output a normalized scalar (e.g., via a structured output like `<score>0.85</score>`), with the RL reward defined as the negative MSE or a ranking-consistent objective against human ratings. Critically, the **stabilizing mechanisms in StableReinforce**—such as Pre-CLIP and Advantage Filtering—remain fully applicable, as they operate on the RL optimization dynamics rather than the specific output semantics.
>
> Thus, while our current experiments focus on the foundational and widely adopted pairwise setting, the underlying algorithmic innovations are **general-purpose enhancements to RL training for MLLMs**, and we believe they can be readily adapted to richer scoring paradigms.
>
> ----
>
> ### [Concern 4] How do the gains evolve as the consistency judge becomes increasingly noisy? Is 0.5 still the optimal weight under noise?
>
> Thank you very much for this excellent and nuanced question—understanding the robustness of our method to noise in the consistency signal is indeed crucial for practical deployment. We sincerely appreciate your attention to this important aspect. To investigate how performance evolves as the consistency judge becomes increasingly noisy, we conducted a series of controlled experiments by varying the quality (and thus reliability) of the consistency judge:
>
> - **Strong judge**: Qwen2.5-VL-7B (the model used in our main experiments)
> - **Weaker judge**: Qwen2.5-VL-3B (a smaller, less capable vision-language model)
> - **Noisy judge**: Qwen2.5-VL-7B with **25% label flipping** on consistency decisions (i.e., with 25% probability, the judge reports the opposite consistency label)
> - **Random judge**: Assigns consistency labels with **50% random chance**, representing extreme noise (no signal)
>
> As noted in our paper, the consistency judgment task itself is relatively simple—determining whether the model’s final choice (“Response 1 is better” or “Response 2 is better”) aligns with the underlying preference label. Consequently, even the smaller model (Qwen2.5-VL-3B) produces only slightly degraded consistency signals, and the optimal consistency weight remains **λ = 0.5**, with only a minor performance drop (71.44% → 70.89% on VL-RewardBench).
>
> However, as noise increases, the optimal weighting does shift. As shown in the table below (evaluated on VL-RewardBench), when we introduce **25% label noise** into the strong judge, the best-performing weight decreases from **0.5 to 0.25**. This indicates that **the model should indeed down-weight the consistency signal as its reliability declines**.
>
> | Consistency Judge                          | Best λ | VL-RewardBench Acc (%) |
> |-------------------------------------------|--------|------------------------|
> | Qwen2.5-VL-7B (clean)                    | 0.5    | 71.44                  |
> | Qwen2.5-VL-3B (weaker but clean)       | 0.5    | 70.89                  |
> | Qwen2.5-VL-7B + 25% label noise          | 0.25   | 67.64                  |
> | Random judge (50% random labels)          | 0.0    | 51.81                  |
>
> These results confirm your intuition: **0.5 is not universally optimal under noise**. In fact, the ideal λ depends on the trustworthiness of the consistency signal. Under extreme noise (random judge), the best strategy is to **disable the consistency reward entirely (λ = 0)**, as any non-zero weight harms performance.
>
> In the revised manuscript, we will include a short discussion (appendix C.11). Thank you again for raising this insightful point—it has helped us better articulate both the strengths and the practical considerations of our consistency-based reward design.

---

> ### Author Response · Authors · 2025-11-26
> **Kindly Request for Reviewer's Feedback**
>
> Dear Reviewer,
>
> Thank you so much for your time in improving our paper!
>
> Since the end of the rebuttal is coming soon, may we know if our response addresses your main concerns? Should you have any further advice, please let us know and we will be more than happy to engage in more discussion and improvements.

---

### Official Review · Reviewer_KA5w · 2025-11-06

**Soundness:** 4
**Presentation:** 4
**Contribution:** 2
**Rating:** 8
**Confidence:** 4

**Summary:**

This paper introduces R1-Reward, a novel Multimodal Reward Model (MRM) trained using a new reinforcement learning (RL) algorithm called StableReinforce. The authors found that directly applying existing RL algorithms like PPO or Reinforce++ to reward modeling leads to extreme training instability and collapse. The instability is traced to flaws in the PPO loss function when handling negative advantages and to extreme outliers created by "Advantage Normalization" in low-variance batches. StableReinforce solves this by introducing "Pre-CLIP," which clamps log probability differences before the exponential function to prevent overflow, and an "Advantage Filter," which removes outliers using the 3-sigma rule. Furthermore, to address inconsistencies where the model's reasoning and final answer contradict each other, the paper introduces a "Consistency Reward" evaluated by an MLLM referee. Trained on a 200K preference dataset , R1-Reward achieves state-of-the-art performance, improving accuracy by 8.4% on the VL Reward-Bench and 14.3% on the Multimodal Reward Bench.

**Strengths:**

The paper's main strength is being the first to successfully use reinforcement learning (RL) to train a multimodal reward model. It cleverly treats the MLLM itself as the reward model, avoiding extra parts like a reward head. The authors showed strong engineering skills by modifying existing RL algorithms to fix critical instability issues that caused them to crash on this task. This practical approach worked very well, allowing their model to achieve SOTA results.

**Weaknesses:**

I think this is a good work. One weakness may be its limited novelty.

**Questions:**

None

---

> ### Author Response · Authors · 2025-11-16
> **Response**
>
> Thank you very much for your thoughtful and encouraging review. We truly appreciate your recognition of our work as the first successful attempt to apply reinforcement learning (RL) to train a multimodal reward model (MRM). Your acknowledgment of our engineering efforts and the strong empirical results means a great deal to us.
>
> We fully agree that our approach builds upon established algorithms such as PPO and Reinforce++. However, the core challenge—and novelty—of our work lies precisely in identifying why these standard algorithms inevitably fail when applied to the novel setting of MLLM-based reward modeling, and in devising principled solutions to overcome those failures. Consequently, StableReinforce (including key components like Pre-CLIP and Advantage Filter) is not merely a set of heuristic engineering tweaks, but rather a targeted algorithmic adaptation essential for making RL practically viable in this new domain. We believe that being the first to stably apply a powerful paradigm like RL to an important yet challenging new problem—MRM training—and to uncover both the root causes of its prior failures and effective remedies, constitutes a significant contribution and a central aspect of our work’s novelty.

---

### Meta-Review · Area_Chair_RgP6 · 2026-01-01

**Summary:**

The paper receives more positive ratings than negative ones. Despite this fact, some positive reviewers also point out that the contribution is limited in theoretical novelty.  Other concerns regarding some technique choices and details, which do not affect the technical contribution, and providing more details on the method may offer a deeper understanding of the method. Overall, this is a good paper for this venue.

**Reviewer Concerns:**

The clarification on some details of the proposed method has been carefully addressed. However, major concerns still exist about how this work contributes to the field, despite introducing RL for the first time in training a multimodal reward model.

**Reviewer Scores:**

The positive reviewers will maintain their positive scores, and reviewer 5DDa may also retain his score, as the rebuttal may not address the core problem regarding the contribution.

---

### Decision · Program_Chairs · 2026-01-26

Accept (Poster)